# A Unified Framework for U-Net Design and Analysis

**Christopher Williams** [*1]    **Fabian Falck** [*,1,2]    **George Deligiannidis** [1]
**Chris Holmes** [1,2]    **Arnaud Doucet** [1]    **Saifuddin Syed** [1]
[1]University of Oxford  [2]The Alan Turing Institute
{williams,fabian.falck,deligian,cholmes, doucet,saifuddin.syed}@stats.ox.ac.uk

## Abstract

U-Nets are a go-to neural architecture across numerous tasks for continuous signals on a square such as images and Partial Differential Equations (PDE), however their design and architecture is understudied. In this paper, we provide a framework for designing and analysing general U-Net architectures. We present theoretical results which characterise the role of the encoder and decoder in a U-Net, their high-resolution scaling limits and their conjugacy to ResNets via preconditioning. We propose Multi-ResNets, U-Nets with a simplified, wavelet-based encoder without learnable parameters. Further, we show how to design novel U-Net architectures which encode function constraints, natural bases, or the geometry of the data. In diffusion models, our framework enables us to identify that high-frequency information is dominated by noise exponentially faster, and show how U-Nets with average pooling exploit this. In our experiments, we demonstrate how Multi-ResNets achieve competitive and often superior performance compared to classical U-Nets in image segmentation, PDE surrogate modelling, and generative modelling with diffusion models. Our U-Net framework paves the way to study the theoretical properties of U-Nets and design natural, scalable neural architectures for a multitude of problems beyond the square.

## 1   Introduction

U-Nets (see Figure 1) are a central architecture in deep learning for continuous signals. Across many tasks as diverse as image segmentation [1, 2, 3, 4, 5], Partial Differential Equation (PDE) surrogate modelling [6, 7] and score-based diffusion models [8, 9, 10, 11, 12], U-Nets are a go-to architecture yielding state-of-the-art performance. In spite of their enormous success, a framework for U-Nets which characterises for instance the specific role of the encoder and decoder in a U-Net or which spaces these operate on is lacking. In this work, we provide such a framework for U-Nets. This allows us to design U-Nets for data beyond a square domain, and enable us to incorporate prior knowledge about a problem, for instance a natural basis, functional constraints, or knowledge about its topology, into the neural architecture.

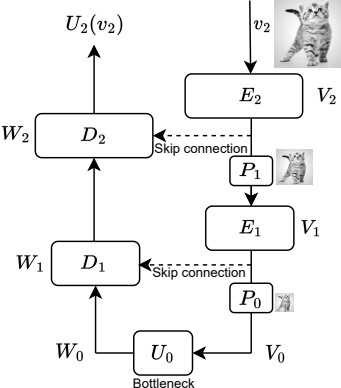

Figure 1: A resolution 2 U-Net (Def. 1). If $E_i = \mathrm{Id}_{V_i}$, this is a Multi-ResNet (see Def. 3).

**The importance of preconditioning.**  We begin by illustrating the importance of the core design principle of U-Nets: *preconditioning*. Preconditioning informally means that initialising an optimisation problem with a 'good' solution greatly benefits learning [13, 14]. Consider a synthetic example using ResNets [15] which are natural in the context of

---

[*]Equal contribution.

37th Conference on Neural Information Processing Systems (NeurIPS 2023).

U-Nets as we will show in §2.3: we are interested in learning a ground-truth mapping $w : V \mapsto W$ and $w(v) = v^2$ over $V = [-1, 1]$ and $W = \mathbb{R}$ using a ResNet $R^{\text{res}}(v) = R^{\text{pre}}(v) + R(v)$ where $R^{\text{pre}}, R : V \mapsto W$. In Figure 2 [Left] we learn a standard ResNet with $R^{\text{pre}}(v) = v$ on a grid of values from $V \times W$, i.e. with inputs $v_i \in V$ and regression labels $w_i = w(v_i) = v_i^2$. In contrast, we train a ResNet with $R^{\text{pre}}(v) = |v|$ [Right] with the same number of parameters and iterations. Both networks have been purposely designed to be weakly expressive (see Appendix B.5 for details). The standard ResNet [Left] makes a poor approximation of the function, whilst the other ResNets [Right] approximation is nearly perfect. This is because $R^{\text{pre}}(v) = |v|$ is a 'good' initial guess or *preconditioner* for $w(v) = v^2$, but $R^{\text{pre}}(v) = v$ is a 'bad' one. This shows the importance of encoding good preconditioning into a neural network architecture and motivates us studying how preconditioning is used in U-Net design.

In this paper, we propose a mathematical framework for designing and analysing general U-Net architectures. We begin with a comprehensive definition of a U-Net which characterises its components and identifies its self-similarity structure which is established via preconditioning. Our theoretical results delineate the role of the encoder and decoder and identify the subspaces they operate on. We then focus on ResNets as natural building blocks for U-Nets that enable flexible preconditioning from lower-resolutions inputs.

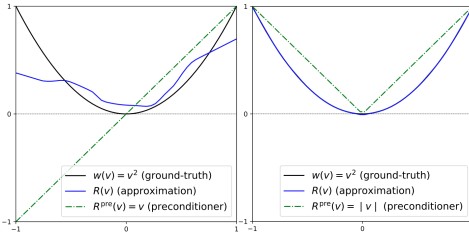

Figure 2: The importance of *preconditioning*.

Our U-Net framework paves the way to designing U-Nets which can model distributions over complicated geometries beyond the square, for instance CW-complexes or manifolds, or diffusions on the sphere [16], without any changes to the diffusion model itself (see Appendix A). It allows us to enforce problem-specific constraints through the architecture, such as boundary conditions of a PDE or a natural basis for a problem. We also analyse why U-Nets with average pooling are a natural inductive bias in diffusion models.

More specifically, our *contributions* are as follows: (a) We provide the first rigorous definition of U-Nets, which enables us to identify their self-similarity structure, high-resolution scaling limits, and conjugacy to ResNets via preconditioning. (b) We present Multi-ResNets, a novel class of U-Nets over a hierarchy of orthogonal wavelet spaces of $L^2(\mathbb{X})$, for compact domain $\mathbb{X}$, with no learnable parameters in its encoder. In our experiments, Multi-ResNets yield competitive and often superior results when compared to a classical U-Net in PDE modelling, image segmentation, and generative modelling with diffusion models. We further show how to encode problem-specific information into a U-Net. In particular, we design U-Nets incorporating a natural basis for a problem which enforces boundary conditions on the elliptic PDE problem, and design and demonstrate proof-of-concept experiments for U-Nets with a Haar wavelet basis over a triangular domain. (c) In the context of diffusion models, we analyse the forward process in a Haar wavelet basis and identify how high-frequency information is dominated by noise exponentially faster than lower-frequency terms. We show how U-Nets with average pooling exploit this observation, explaining their go-to usage.

## 2 U-Nets: Neural networks via subspace preconditioning

The goal of this section is to develop a mathematical framework for U-Nets which introduces the fundamental principles that underpin its architecture and enables us to design general U-Net architectures. All theoretical results are proven in Appendix A. We commence by defining the U-Net.

### 2.1 Anatomy of a U-Net

**Definition 1. U-Net.** Let $V$ and $W$ be measurable spaces. A *U-Net* $\mathcal{U} = (\mathcal{V}, \mathcal{W}, \mathcal{E}, \mathcal{D}, \mathcal{P}, U_0)$ comprises six components:

1. *Encoder subspaces:* $\mathcal{V} = (V_i)_{i=0}^{\infty}$ are nested subsets of $V$ such that $\lim_{i \to \infty} V_i = V$.
2. *Decoder subspaces:* $\mathcal{W} = (W_i)_{i=0}^{\infty}$ are nested subsets of $W$ such that $\lim_{i \to \infty} W_i = W$.
3. *Encoder operators:* $\mathcal{E} = (E_i)_{i=1}^{\infty}$ where $E_i : V_i \mapsto V_i$ denoted $E_i(v_i) = \tilde{v}_i$.
4. *Decoders operators:* $\mathcal{D} = (D_i)_{i=1}^{\infty}$ where $D_i : W_{i-1} \times V_i \mapsto W_i$ at resolution $i$ denoted $D_i(w_{i-1}|v_i)$. The $v_i$ component is called the *skip connection*.

5. *Projection operators:* $\mathcal{P} = (P_i)_{i=0}^{\infty}$, where $P_i : V \mapsto V_i$, such that $P_i(v_i) = v_i$ for $v_i \in V_i$.
6. *Bottleneck*: $U_0$ is the mapping $U_0 : V_0 \mapsto W_0$, enabling a compressed representation of the input.

The *U-Net of resolution* $i$ is the mapping $U_i : V_i \mapsto W_i$ defined through the recursion (see Figure 3):

$$U_i(v_i) = D_i(U_{i-1}(P_{i-1}(\tilde{v}_i))|\tilde{v}_i), \qquad i = 1, 2, \ldots. \tag{1}$$

We illustrate the Definition of a U-Net in Figure 1. Definition 1 includes a wide array of commonly used U-Net architectures, from the seminal U-Net [1], through to modern adaptations used in large-scale diffusion models [8, 9, 10, 11, 12], operator learning U-Nets for PDE modelling [6, 7], and custom designs on the sphere or graphs [17, 18]. Our framework also comprises models with multiple channels (for instance RGB in images) by choosing $V_i$ and $W_i$ for a given resolution $i$ to be product spaces $V_i = \bigotimes_{k=1}^{M} V_{i,k}$ and $W_i = \bigotimes_{k=1}^{M} W_{i,k}$ for $M$ channels when necessary[2]. Remarkably, despite their widespread use, to the best of our knowledge, our work presents the first formal definition of a U-Net. Definition 1 not only expands the scope of U-Nets beyond

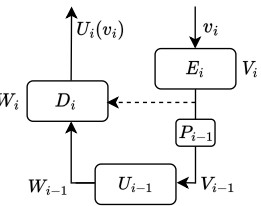

Figure 3: Recursive structure of a U-Net.

problems confined to squared domains, but also naturally incorporates problem-specific information such as a natural basis, boundary conditions or topological structure as we will show in §3.2 and 3.3. This paves the way for designing inherently scalable neural network architectures capable of handling complicated geometries, for instance manifolds or CW-complexes. In the remainder of the section, we discuss and characterise the components of a U-Net.

**Encoder and decoder subspaces.** We begin with the spaces in $V$ and $W$ which a U-Net acts on. Current literature views U-Nets as learnable mappings between input and output tensors. In contrast, this work views U-Nets and their encoder and decoder as operators on spaces that must be chosen to suit our task. In order to perform computations in practice, we must restrict ourselves to subspaces $V_i$ and $W_i$ of the potentially infinite-dimensional spaces $V$ and $W$. For instance, if our U-Net is modelling a mapping between images, which can be viewed as bounded functions over a squared domain $\mathbb{X} = [0, 1]^2$, it is convenient to choose $V$ and $W$ as subspaces of $L^2(\mathbb{X})$, the space of square-integrable functions [19, 20] (see Appendix C). Here, a data point $w_i$ in the decoder space $W_i$ is represented by the coefficients $c_{i,j}$ where $w_i = \sum_j c_{i,j} e_{i,j}$ and $\{e_{i,j}\}_j$ is a (potentially orthogonal) basis for $W_i$. We will consider precisely this case in §3.1. The projected images on the subspace $V_i, W_i$ are still functions, but piece-wise constant ones, and we store the values these functions obtain as 'pixel' tensors in our computer [20] (see Figure 22 in Appendix A).

**Role of encoder, decoder, projection operators.** In spite of their seemingly symmetric nature and in contrast to common understanding, the roles of the encoder and decoder in a U-Net are fundamentally different from each other. The decoder on resolution $i$ learns the transition from $W_{i-1}$ to $W_i$ while incorporating information from the encoder on $V_i$ via the skip connection. The encoder $E_i$ can be viewed as a change of basis mapping on the input space $V_i$ at resolution $i$ and is not directly tied to the approximation the decoder makes on $W_i$. This learned change of basis facilitates the decoder's approximation on $W_i$. In §3.1, we will further extend our understanding of the encoder and discuss its implications for designing U-Nets. The projection operators serve to extract a compressed input. They are selected to suit task-specific needs, such as pooling operations (e.g. average pooling, max pooling) in the context of images, or orthogonal projections if $V$ is a Hilbert space. Note that there is no embedding operator[3], the operator facilitating the transition to a higher-resolution space, explicitly defined as it is invariably the natural inclusion of $W_{i-1}$ into $W_i$.

**Self-similarity of U-Nets via preconditioning.** The key design principle of a U-Net is preconditioning. The U-Net of resolution $i - 1$, $U_{i-1}$ makes an approximation on $W_{i-1}$ which is input of and preconditions $U_i$. Preconditioning facilitates the transition from $W_{i-1}$ to $W_i$ for the decoder. In the encoder, the output of $P_{i-1}E_i$ is the input of $U_{i-1}$. When our underlying geometry is refinable (such as a square), we may use a refinable set of basis functions. In the standard case of a U-Net with average pooling on a square domain, our underlying set of basis functions are (Haar) wavelets (see §2.3) – refinable basis functions defined on a refinable geometry. This preconditioned design of

---

[2]Implementation wise this corresponds to taking the Kronecker product across the spaces $V_{i,k}$ and $W_{i,k}$, respectively.

[3]In practice, if we for instance learn a transposed convolution operation, it can be equivalently expressed as a standard convolution operation composed with the natural inclusion operation.

U-Nets reveals a self-similarity structure (see Figure 3) inherent to the U-Net when the underlying space has a refinable geometry. This enables both an efficient multi-resolution approximation of U-Nets [20] and makes them modular, as a U-Net on resolution $i - 1$ is a coarse approximation for a U-Net on resolution $i$. We formalise this notion of preconditioning in Proposition 1 in §2.3.

## 2.2 High-resolution scaling behavior of U-Nets

Given equation (1), it is clear that the expressiveness of a U-Net $\mathcal{U}$ is governed by the expressiveness of its decoder operators $\mathcal{D}$. If each $D_i$ is a universal approximator [21], then the corresponding U-Nets $U_i$ likewise have this property. Assuming we can represent any mapping $U_i : V_i \mapsto W_i$ as a U-Net, our goal now is to comprehend the role of increasing the resolution in the design of $\mathcal{U}$, and to discern whether any function $U : V \to W$ can be represented as a high-resolution limit $\lim_{i \to \infty} U_i$ of a U-Net. We will explore this question in the context of regression problems of increasing resolution.

To obtain a tractable answer, we focus on choosing $W$ as a *Hilbert space*, that is $W$ equipped with an inner product. This allows us to define $\mathcal{W}$ as an increasing sequence of *orthogonal* subspaces of $W$. Possible candidates for orthogonal bases include certain Fourier frequencies, wavelets (of a given order) or radial basis functions. The question of which basis is optimal depends on our problem and data at hand: some problems may be hard in one basis, but easy in another. In §4, Haar wavelets are a convenient choice. Let us define $\mathcal{S}$ as infinite resolution data in $V \times W$ and $\mathcal{S}_i$ as the finite resolution projection in $V_i \times W_i$ comprising of $(v_i, w_i) = (P_i(v), Q_i(w))$ for each $(v, w) \in \mathcal{S}$. Here, $P_i$ is the U-Net projection onto $V_i$, and $Q_i : W \mapsto W_i$ is the orthogonal projection onto $W_i$. Assume $U_i^*$ and $U^*$ are solutions to the finite and infinite resolution regression problems $\min_{U_i \in \mathcal{F}_i} \mathcal{L}_i(U_i | \mathcal{S}_i)$ and $\min_{U \in \mathcal{F}} \mathcal{L}(U | \mathcal{S})$ respectively, where $\mathcal{F}_i$ and $\mathcal{F}$ represent the sets of measurable functions mapping $V_i \mapsto W_i$ and $V \mapsto W$. Let $\mathcal{L}_i$ and $\mathcal{L}$ denote the $L^2$ losses:

$$\mathcal{L}_i(U_i | \mathcal{S}_i)^2 := \frac{1}{|\mathcal{S}_i|} \sum_{(w_i, v_i) \in \mathcal{S}_i} \|w_i - U_i(v_i)\|^2, \quad \mathcal{L}(U | \mathcal{S})^2 := \frac{1}{|\mathcal{S}|} \sum_{(w, v) \in \mathcal{S}} \|w - U(v)\|^2.$$

The following result analyses the relationship between $U_i^*$ and $U^*$ as $i \to \infty$ where $\mathcal{L}_{i|j}$ and $\mathcal{L}_{|j}$ are the losses above conditioned on resolution $j$ (see Appendix A).

**Theorem 1.** *Suppose $U_i^*$ and $U^*$ are solutions of the $L^2$ regression problems above. Then, $\mathcal{L}_{i|j}(U_i^*) \leq \mathcal{L}_{|j}(U^*)$ with equality as $i \to \infty$. Further, if $Q_i U^*$ is $V_i$-measurable, then $U_i^* = Q_i U^*$ minimises $\mathcal{L}_i$.*

Theorem 1 states that solutions of the finite resolution regression problem converge to solutions of the infinite resolution problem. It also informs us how to choose $V_i$ relative to $W_i$. If we have a $\mathcal{V}$ where for the decoders on $\mathcal{W}$, the $W_i$ component of $U^*$, $Q_i U^*$, relies solely on the input up to resolution $i$, then the prediction from the infinite-dimensional U-Net projected to $W_i$ can be made by the U-Net of resolution $i$. The optimal choice of $V_i$ must be expressive enough to encode the information necessary to learn the $W_i$ component of $U^*$. This suggests that if $V_i$ lacks expressiveness, we risk efficiency in learning the optimal value of $U_i^*$. However, if $V_i$ is too expressive, no additional information is gained, and we waste computational effort. Therefore, we should choose $V_i$ to encode the necessary information for learning information on resolution $i$. For example, when modelling images, if we are interested in low-resolution features on $W_i$, high-resolution information is extraneous, as we will further explore in §4 in the context of diffusion models.

## 2.3 U-Nets are conjugate to ResNets

Next, our goal is to understand why ResNets are a natural choice in U-Nets. We will uncover a conjugacy between U-Nets and ResNets. We begin by formalising ResNets in the context of U-Nets.

**Definition 2. ResNet, Residual U-Nets.** Given a measurable space $X$ and a vector space $Y$, a mapping $R : X \to Y$ is defined as a *ResNet* preconditioned on $R^{\mathrm{pre}} : X \to Y$ if $R(x) = R^{\mathrm{pre}}(x) + R^{\mathrm{res}}(x)$, where $R^{\mathrm{res}}(x) = R(x) - R^{\mathrm{pre}}(x)$ is the *residual* of $R$. A *Residual U-Net* is a U-Net $\mathcal{U}$ where $\mathcal{W}, \mathcal{V}$ are sequences of vector spaces, and the encoder and decoder operators $\mathcal{E}, \mathcal{D}$ are ResNets preconditioned on $E_i^{\mathrm{pre}}(v_i) = v_i$ and $D_i^{\mathrm{pre}}(w_{i-1} | v_i) = w_{i-1}$, respectively.

A preconditioner initialises a ResNet, then the ResNet learns the residual relative to it. The difficulty of training a ResNet scales with the deviation from the preconditioner, as we saw in our synthetic experiment in §1. In U-Nets, ResNets commonly serve as encoder and decoder operators. In encoders,

preconditioning on the identity on $V_i$ allow the residual encoder $E_i^{\text{res}}$ to learn a change of basis for $V_i$, which we will discuss in more detail in §3.1. In decoders, preconditioning on the identity on the lower resolution subspace $W_{i-1}$ allows the residual decoder $D_i^{\text{res}}$ to learn from the lower resolution and the skip connection. Importantly, ResNets can compose to form new ResNets, which, combined with the recursion (1), implies that residual U-Nets are conjugate to ResNets.

**Proposition 1.** *If $\mathcal{U}$ is a residual U-Net, then $U_i$ is a ResNet preconditioned on $U_i^{\text{pre}}(v_i) = U_{i-1}(\tilde{v}_{i-1})$, where $\tilde{v}_{i-1} = P_{i-1}(E_i(v_i))$.*

Proposition 1 states that a U-Net at resolution $i$ is a ResNet preconditioned on a U-Net of lower resolution. This suggests that $U_i^{\text{res}}$ learns the information arising from the resolution increase. We will discuss the specific case $E_i = \text{Id}_{V_i}$, and $U^{\text{pre}}(v_i) = U_{i-1}(P_{i-1}(v_i))$ in §3.1. Proposition 1 also enables us to interpret $\lim_{i \to \infty} U_i$ from §2.2 as a ResNet's 'high-resolution' scaling limit. This is a new scaling regime for ResNets, different to time scaled Neural ODEs [22], and warrants further exploration in future work. Finally, we provide an example of the common *Residual U-Net*.

**Example 1. Haar Wavelet Residual U-Net.** A *Haar wavelet residual U-Net* $\mathcal{U}$ is a residual U-Net where: $V = W = L^2(\mathbb{X})$, $V_i = W_i$ are multi-resolution Haar wavelet spaces (see Appendix C.3), and $P_i = \text{proj}_{V_i}$ is the orthogonal projection.

We design Example 1 with images in mind, noting that similar data over a squared domain such as PDE (see §5) are also applicable to this architecture. We hence choose Haar wavelet [23] subspaces of $L^2(\mathbb{X})$, the space of square integrable functions and a Hilbert space, and use average pooling as the projection operation $P_i$ [20]. Haar wavelets will in particular be useful to analyse why U-Nets are a good inductive bias in diffusion models (see §4). U-Net design and their connection to wavelets has also been studied in [24, 25, 26].

## 3 Generalised U-Net design

In this section, we provide examples of different problems for which our framework can define a natural U-Net architecture. Inspired by Galerkin subspace methods [27], our goal is to use our framework to generalise the design of U-Nets beyond images over a square. Our framework also enables us to encode problem-specific information into the U-Net, such as a natural basis or boundary conditions, which it no longer needs to learn from data, making the U-Net model more efficient.

### 3.1 Multi-ResNets

**A closer look at the U-Net encoder.** To characterise a U-Net in Definition 1, we must in particular choose the encoder subspaces $\mathcal{V}$. This choice depends on our problem at hand: for instance, if the inputs are images, choosing Haar wavelet subspaces is most likely favourable, because we can represent and compress images in a Haar wavelet basis well, noting that more complex (orthogonal) wavelet bases are possible [28, 29]. What if we choose $\mathcal{V}$ unfavourably? This is where the encoder comes in. While the encoder subspaces $\mathcal{V}$ define an initial basis for our problem, the encoder learns a *change of basis* map to a new, implicit basis $\tilde{v}_i = E_i(v_i)$ which is more favourable. This immediately follows from Eq. (1) since $U_i$ acts on $V_i$ through $\tilde{v}_i$. The initial subspaces $\mathcal{V}$ can hence be viewed as a prior for the input compression task which the encoder performs.

Given our initial choice of the encoder subspaces $\mathcal{V}$, the question whether and how much work the encoders $\mathcal{E}$ have to do depends on how far away our choice is from the optimal choice $\tilde{\mathcal{V}}$ for our problem. This explains why the encoders $E_i$ are commonly chosen to be ResNets preconditioned on the identity $E_i^{\text{pre}} = \text{Id}_{V_i}$, allowing the residual encoder $E_i^{\text{res}}$ to learn a change of basis. If we had chosen the optimal sequence of encoder subspaces, the residual operator would not have to do any work; leaving the encoder equal to the precondition $E_i = \text{Id}_{V_i}$. It also explains why in practice, encoders are in some cases chosen significantly smaller than the decoder [30], as a ResNet encoder need not do much work given a good initial choice of $\mathcal{V}$. It is precisely this intuition which motivates our second example of a U-Net, the Multi-Res(olution) Res(idual) Network (*Multi-ResNet*).

**Definition 3. Multi-ResNets.** A Multi-ResNet is a residual U-Net with encoder $E_i = \text{Id}_{V_i}$.

**Example 2. Haar Wavelet Multi-ResNets.** A Haar Wavelet Multi-ResNet is a Haar Wavelet Residual U-Net with encoder $E_i = \text{Id}_{V_i}$.

We illustrate the Multi-ResNet, a novel U-Net architecture, in Figure 1 where we choose $E_i = \mathrm{Id}_{V_i}$. Practically speaking, the Multi-ResNet simplifies its encoder to have no learnable parameters, and simply projects to $V_i$ on resolution $i$. The latter can for the example of Haar wavelets be realised by computing a multi-level Discrete Wavelet Transform (DWT) (or equivalently average pooling) over the input data [20]. Multi-ResNets allow us to save the parameters in the encoder, and instead direct them to bolster the decoder. In our experiments in §5.1, we compare Multi-ResNets to Residual U-Nets and find that for PDE surrogate modelling and image segmentation, Multi-ResNets yield superior performance to Residual U-Nets as Haar wavelets are apparently a good choice for $\mathcal{V}$, while for other problems, choosing Haar wavelets is suboptimal. Future work should hence investigate how to optimally choose $\mathcal{V}$ for a problem at hand. To this end, we will discuss natural bases for $\mathcal{V}$ and $\mathcal{W}$ for specific problems in the remainder of this section.

### 3.2 U-Nets which guarantee boundary conditions

Next, our main goal is to show how to design U-Nets which choose $\mathcal{W}$ in order to encode constraints on the output space directly into the U-Net architecture. This renders the U-Net more efficient as it no longer needs to learn the constraints from data. We consider an example from *PDE surrogate modelling*, approximating solutions to PDE using neural networks, a nascent research direction where U-Nets already play an important role [6], where our constraints are given boundary conditions and the solution space of our PDE. In the *elliptic boundary value problem* on $\mathbb{X} = [0,1]$ [31], the task is to predict a weak (see Appendix A) PDE solution $u$ from its forcing term $f$ given by

$$\Delta u = f, \qquad\qquad u(0) = u(1) = 0, \qquad\qquad (2)$$

where $u$ is once weakly differentiable when the equation is viewed in its weak form, $f \in L^2(\mathbb{X})$ and $\Delta u$ is the Laplacian of $u$. In contrast to Examples 1 and 2, we choose the decoder spaces as subspaces of $W = \mathcal{H}_0^1$, the space of one weakly differentiable functions with nullified boundary condition, a Hilbert space (see Appendix A), and choose $V = L^2(\mathbb{X})$, the space of square integrable functions. This choice ensures that input and output functions of our U-Net are in the correct function class for the prescribed problem. We now want to choose a basis to construct the subspaces $\mathcal{V}$ and $\mathcal{W}$ of $V$ and $W$. For $\mathcal{V}$, just like in Multi-ResNets in Example 2, we choose $V_j$ to be the Haar wavelet space of resolution $j$, an orthogonal basis. For $\mathcal{W}$, we also choose a refinable basis, but one which is natural to $\mathcal{H}_0^1$. In particular, we choose $W_i = \mathrm{span}\{\phi_{k,j} : j \le k, k = 1, \ldots, 2^{i-1}\}$ where

$$\phi_{k,j}(x) = \phi(2^k x + j/2^k), \qquad \phi(x) = 2x \cdot 1_{[0,1/2)}(x) + (2 - 2x) \cdot 1_{[1/2,1]}(x). \qquad (3)$$

This constructs an orthogonal basis of $\mathcal{H}_0^1$, illustrated in Figure 4, which emulates our design choice in Section 3.1 where the orthogonal Haar wavelet basis was beneficial as $W$ was $L^2$-valued. Each $\phi_{k,j}$ obeys the regularity and boundary conditions of our PDE, and consequently, an approximate solution from our U-Net obeys these functional constraints as well.

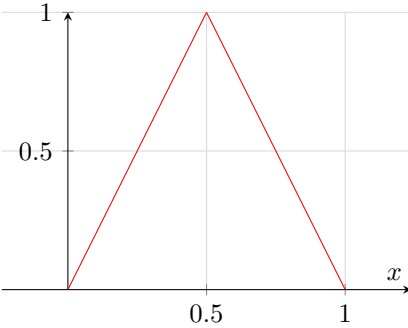
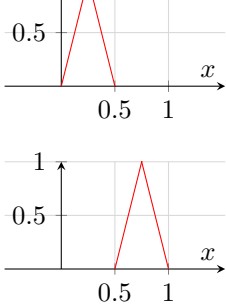

Figure 4: Refinement of an orthogonal basis for $\mathcal{H}_0^1 = \mathrm{span}\{\phi_{0,0}, \phi_{1,0}, \phi_{1,1}\}$. We visualise the graphs of basis functions defined in (3): [Left] $\phi_{0,0} = \phi$, [Top Right] $\phi_{1,0}$, and [Bottom Right] $\phi_{1,1}$. When increasing resolution, steeper triangular-shaped basis functions are constructed.

These constraints are encoded into the U-Net architecture and hence need not be learned from data. This generalised U-Net design paves the way to broaden the application of U-Nets, analogous to the choice of bases for Finite Element [32] or Discontinuous Galerkin methods [33].

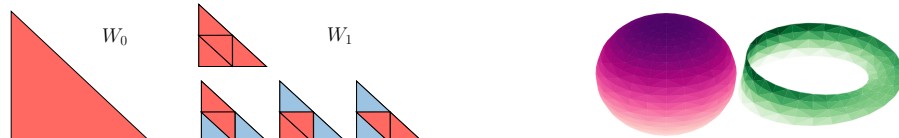

Figure 5: U-Nets encoding the topological structure of a problem. [Left] A refinable Haar wavelet basis with basis functions on a right triangle, $\phi_{i,j=0} = \mathbb{1}_{\text{red}} - \mathbb{1}_{\text{blue}}$. [Right] A sphere and a Möbius strip meshed with a Delaunay triangulation [35, 36]. Figures and code as modified from [37].

### 3.3 U-Nets for complicated geometries

Our framework further allows us to design U-Nets which encode the geometry of the input space right into the architecture. This no longer requires to learn the geometric structure from data and enables U-Nets for complicated geometries. In particular, we are motivated by *tessellations*, the partitioning of a surface into smaller shapes, which play a vital role in modelling complicated geometries across a wide range of engineering disciplines [34]. We here focus on U-Nets on a triangle due to the ubiquitous use of triangulations, for instance in CAD models or simulations, but note that our design principles can be applied to other shapes featuring a self-similarity property. We again are interested in finding a natural basis for this geometry, and characterise key components of our U-Net.

In this example, neither $W$ nor $V$ are selected to be $L^2(\mathbb{X})$ valued on the unit square (or rectangle) $\mathbb{X}$. Instead, in contrast to classical U-Net design in literature, $W = V = L^2(\triangle)$, where $\triangle$ is a right-triangular domain illustrated in Figure 5 [Left]. Note that this right triangle has a self-similarity structure in that it can be constructed from four smaller right triangles, continuing recursively. A refinable Haar wavelet basis for this space can be constructed by starting from $\phi_{i,0} = \mathbb{1}_{\text{red}} - \mathbb{1}_{\text{blue}}$ for $j = 0$ as illustrated in Figure 5 [Left]. This basis can be refined through its self-similarity structure to define each subspace from these basis functions via $W_i = V_i = \text{Span}\{\phi_{k,j} : j \leq k, k = 1, \dots, 2^{i-1}\}$ (see Appendix A for details). In §5.2, we investigate this U-Net design experimentally. In Appendix A we sketch out how developing our U-Net on triangulated manifolds enables score-based diffusion models on a sphere [16] without any adjustments to the diffusion process itself. This approach can be extended to complicated geometries such as manifolds or CW-complexes as illustrated in Figure 5 [Right].

## 4 Why U-Nets are a useful inductive bias in diffusion models

U-Nets are the go-to neural architecture for diffusion models particularly on image data, as demonstrated in an abundance of previous work [8, 9, 10, 11, 12, 38, 39, 40, 41]. However, the reason why U-Nets are particularly effective in the context of diffusion models is understudied. Our U-Net framework enables us to analyse this question. We focus on U-Nets over nested Haar wavelet subspaces $\mathcal{V} = \mathcal{W}$ that increase to $V = W = L^2([0, 1])$, with orthogonal projection $Q_j : W \mapsto W_i$ on to $W_i$ corresponding to an average pooling operation $Q_i$ [42] (see Appendix C). U-Nets with average pooling are a common choice for diffusion models in practice, for instance when modelling images [8, 9, 38, 39, 40]. We provide theoretical results which identify that high-frequencies in a forward diffusion process are dominated by noise exponentially faster, and how U-Nets with average pooling exploit this in their design.

Let $X \in W$ be an infinite resolution image. For each resolution $i$ define the image $X_i = Q_i X \in W_i$ on $2^i$ pixels which can be described by $X_i = \sum_k X_i^{(k)} \phi_k$, where $\Phi = \{\phi_k : k = 1, \dots, 2^i\}$ is the standard (or 'pixel') basis. The image $X_i$ is a projection of the infinite resolution image $X$ to the finite resolution $i$. We consider the family of denoising processes $\{X_i(t)\}_{i=1}^{\infty}$, where for resolution $i$, the process $X_i(t) = \sum_k X_i^{(k)}(t)\phi_k \in W_i$ is initialised at $X_i$ and evolves according to the denoising diffusion forward process (DDPM, [8]) at each pixel $X_i^{(k)}(t) := \sqrt{1 - \alpha_t} X_i^{(k)} + \sqrt{\alpha_t}\varepsilon^{(k)}$ for standard Gaussian noise $\varepsilon^{(k)}$. We now provide our main result (see Appendix A for technical details).

**Theorem 2.** *For time $t \geq 0$ and $j \geq i$, $Q_i X_j(t) \overset{d}{=} X_i(t)$. Furthermore if $X_i(t) = \sum_{j=0}^{i} \widehat{X}^{(j)}(t) \cdot \widehat{\phi}_j$, be the decomposition of $X_i(t)$ in its Haar wavelet frequencies (see Appendix C). Each component $\widehat{X}^{(j)}(t)$ of the vector has variance $2^{j-1}$ relative to the variance of the base Haar wavelet frequency.*

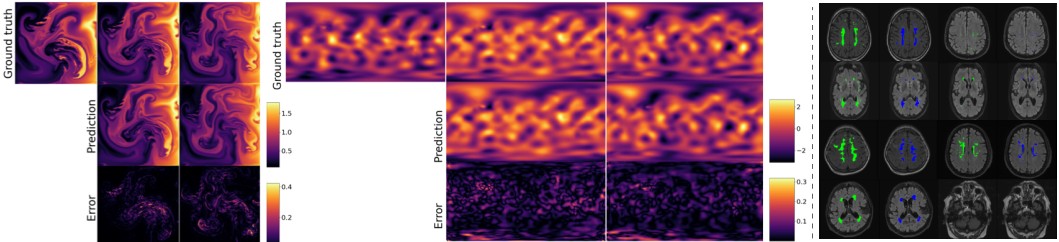

Figure 6: PDE modelling and image segmentation with a Multi-ResNet. [Left,Middle] Rolled out PDE trajectories (ground-truth, prediction, $L^2$-error) from the `Navier-Stokes` [Left], and the `Shallow Water` equation [Middle]. Figure and code as modified from [6, Figure 1]. [Right] MRI images from `WMH` with overlayed ground-truth (green) and prediction (blue) mask.

Theorem 2 analyses the noising effect of a forward diffusion process in a Haar wavelet basis. It states that the noise introduced by the forward diffusion process is more prominent in the higher-frequency wavelet coefficients (large $k$), whereas the lower-frequency coefficients (small $k$) preserves the signal. Optimal recovery of the signal in such scenario has been investigated in [43], where soft thresholding of the wavelet coefficients provides a good $L^2$ estimator of the signal and separates this from the data. In other words, if we add i.i.d. noise to an image, we are noising the higher frequencies faster than the lower frequencies. In particular, high frequencies are dominated by noise *exponentially* faster. It also states that the noising effect of our diffusion on resolution $i$, compared to the variance of the base frequency $i = 0$, blows up the higher-frequency details as $i \to \infty$ for any positive diffusion time.

We postulate that U-Nets with average pooling exploit precisely this observation. Recall that the primary objective of a U-Net in denoising diffusion models is to separate the signal from the noise which allows reversing the noising process. In the 'learning noise' or $\epsilon$ recovery regime, the network primarily distinguishes the noise from the signal in the input. Yet, our analysis remains relevant, as it fundamentally pertains to the signal-to-noise ratio. Through average pooling, the U-Net discards those higher-frequency subspaces which are dominated by noise, because average pooling is conjugate to projection in a Haar wavelet basis [20, Theorem 2]. This inductive bias enables the encoder and decoder networks to focus on the signal on a low enough frequency which is not dominated by noise. As the subspaces are coupled via preconditioning, the U-Net can learn the signal which is no longer dominated by noise, added on each new subspace. This renders U-Nets a computationally efficient choice in diffusion models and explains their ubiquitous use in this field.

## 5 Experiments

We conduct three main experimental analyses: (A) Multi-ResNets which feature an encoder with no learnable parameters as an alternative to classical Residual U-Nets, (B) Multi-resolution training and sampling, (C) U-Nets encoding the topological structure of triangular data. We refer to Appendix B.4 for our Ablation Studies, where a key result is that U-Nets crucially benefit from the skip connections, hence the encoder is successful and important in compressing information. We also analyse the multi-resolution structure in U-Nets, and investigate different orthogonal wavelet bases. These analyses are supported by experiments on three tasks: (1) Generative modelling of images with diffusion models, (2) PDE Modelling, and (3) Image segmentation. We choose these tasks as U-Nets are a go-to and competitive architecture for them. We report the following performance metrics with mean and standard deviation over three random seeds on the test set: FID score [44] for (1), rollout mean-squared error (r-MSE) [6] for (2), and the Sørensen–Dice coefficient (Dice) [45, 46] for (3). As datasets, we use `MNIST` [47], a custom triangular version of MNIST (`MNIST-Triangular`) and `CIFAR10` [48] for (1), `Navier-stokes` and `Shallow water` equations [49] for (2), and the MICCAI 2017 White Matter Hyperintensity (`WMH`) segmentation challenge dataset [50, 51] for (3). We provide our PyTorch code base at https://github.com/FabianFalck/unet-design. We refer to Appendices B, and D for details on experiments, further experimental results, the datasets, and computational resources used.

Table 1: Quantitative performance of the (Haar wavelet) Multi-ResNet compared to a classical (Haar wavelet) Residual U-Net on two PDE modelling and an image segmentation task.

| Dataset | Neural architecture | # Params. | r-MSE ↓ / Dice ↑ |
|---|---|---|---|
| **Navier-stokes** $128 \times 128$ | Residual U-Net | 34.5 M | $0.0057 \pm 2 \cdot 10^{-5}$ |
| | Multi-ResNet, no params. added in dec. (*ours*) | 15.7 M | $0.0107 \pm 9 \cdot 10^{-5}$ |
| | Multi-ResNet, saved params. added in dec. (*ours*) | 34.5 M | $\mathbf{0.0040 \pm 2 \cdot 10^{-5}}$ |
| **Shallow water** $96 \times 192$ | Residual U-Net | 34.5 M | $0.1712 \pm 0.0005$ |
| | Multi-ResNet, no params. added in dec. (*ours*) | 15.7 M | $0.4899 \pm 0.0156$ |
| | Multi-ResNet, saved params. added in dec. (*ours*) | 34.5 M | $\mathbf{0.1493 \pm 0.0070}$ |
| **WMH** $200 \times 200$ | Residual U-Net | 2.2 M | $0.8069 \pm 0.0234$ |
| | Multi-ResNet, no params. added in dec. (*ours*) | 1.0 M | $0.8190 \pm 0.0047$ |
| | Multi-ResNet, saved params. added in dec. (*ours*) | 2.2 M | $\mathbf{0.8346 \pm 0.0388}$ |

## 5.1 The role of the encoder in a U-Net

In §3.1 we motivated Multi-ResNets, Residual U-Nets with identity operators as encoders over Haar wavelet subspaces $\mathcal{V} = \mathcal{W}$ of $V = W = L^2(\mathbb{X})$. We analysed the role of the encoder as learning a change of basis map and found that it does not need to do any work, if $\mathcal{V}$, the initial basis, is chosen optimally for the problem at hand. Here, we put exactly this hypothesis derived from our theory to a test. We compare classical (Haar wavelet) Residual U-Nets with a (Haar wavelet) Multi-ResNet. In Table 1, we present our results quantified on trajectories from the Navier-stokes and Shallow water PDE equations unrolled over several time steps and image segmentation as illustrated in Figure 6. Our results show that Multi-ResNets have competitive and sometimes superior performance when compared to a classical U-Net with roughly the same number of parameters. Multi-ResNets outperform classical U-Nets by 29.8%, 12.8% and 3.4% on average over three random seeds, respectively. In Appendix B.1, we also show that U-Nets outperform FNO [52], another competitive architecture for PDE modelling, in this experimental setting.

CIFAR10

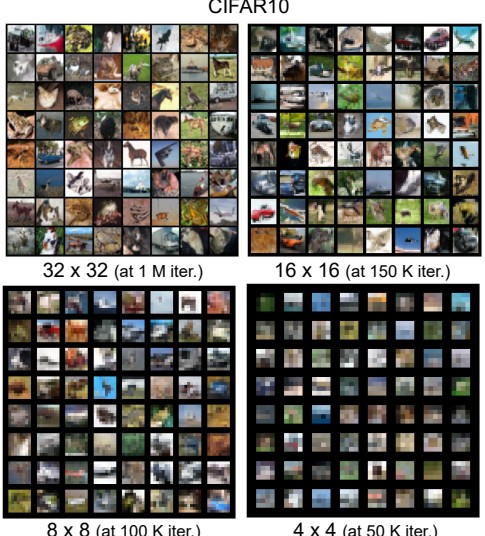

32 x 32 (at 1 M iter.)   16 x 16 (at 150 K iter.)

8 x 8 (at 100 K iter.)   4 x 4 (at 50 K iter.)

Figure 7: Preconditioning enables multi-resolution training and sampling of diffusion models.

For the practitioner, this is a rather surprising result. We can simplify classical U-Nets by replacing their parameterised encoder with a fixed, carefully chosen hierarchy of linear transformation as projection operators $P_i$, here a multi-level Discrete Wavelet Transform (DWT) using Haar wavelets, and identity operators for $E_i$. This 'DWT encoder' has no learnable parameters and comes at almost no computational cost. We then add the parameters we save into the decoder and achieve competitive and—on certain problems—strictly better performance when compared with a classical U-Net. However, as we show for generative modelling with diffusion models in Appendix B.1, Multi-ResNets are competitive with, yet inferior to Residual U-Nets, because the initial basis which $\mathcal{V}$ imposes is suboptimal and the encoder would benefit from learning a better basis. This demonstrates the strength of our framework in understanding the role of the encoder and when it is useful to parameterise it, which depends on our problem at hand. It is now obvious that future empirical work should explore how to choose $\mathcal{V}$ (and $\mathcal{W}$) optimally, possibly eliminating the need of a parameterised encoder, and also carefully explore how to optimally allocate and make use of the (saved) parameters in Multi-ResNets [53, 54].

## 5.2 Staged training enables multi-resolution training and inference

Having characterised the self-similarity structure of U-Nets in §2, a natural idea is to explicitly train the U-Net $\mathcal{U}$ on resolution $i-1$ first, then train the U-Net on resolution $i$ while preconditioning on $U_{i-1}$, continuing recursively. Optionally, we can freeze the weights of $\mathcal{U}_{i-1}$ upon training on resolution $i-1$. We formalise this idea in Algorithm 1. Algorithm 1 enables training and inference of U-Nets on multiple resolutions, for instance when several datasets are available. It has two additional advantages. First, it makes the U-Net modular with respect to data. When data on a higher-resolution is available, we can reuse our U-Net pretrained on a lower-resolution in a principled way. Second, it enables to checkpoint the model during prototyping and experimenting with the model as we can see low-resolution outputs of our U-Net early.



Figure 8: U-Nets encode the geometric structure of data.

In Figure 7 we illustrate samples from a DDPM diffusion model [8] with a Residual U-Net trained with Algorithm 1. We use images and noise targets on multiple resolutions (CIFAR10/MNIST: $\{4 \times 4, 8 \times 8, 16 \times 16, 32 \times 32\}$) as inputs during each training stage. We observe high-fidelity samples on multiple resolutions at the end of each training stage, demonstrating how a U-Net trained via Algorithm 1 can utilise the data available on four different resolutions. It is also worth noting that training with Algorithm 1 as opposed to single-stage training (as is standard practice) does not substantially harm performance of the highest-resolution samples: (FID on CIFAR10: staged training: $8.33 \pm 0.010$; non-staged training: $7.858 \pm 0.250$). We present results on MNIST, Navier-Stokes and Shallow water, with Multi-ResNets, and with a strict version of Algorithm 1 where we freeze $E_i^\theta$ and $D_i^\theta$ after training on resolution $i$ in Appendix B.2.

### 5.3 U-Nets encoding topological structure

In §3.3, we showed how to design U-Nets with a natural basis on a triangular domain, which encodes the topological structure of a problem into its architecture. Here, we provide proof-of-concept results for this U-Net design. In Figure 8, we illustrate samples from a DDPM diffusion model [8] with a U-Net where we choose $\mathcal{V}$ and $\mathcal{W}$ as Haar wavelet subspaces of $W = V = L^2(\triangle)$ (see §3.3), ResNet encoders and decoders and average pooling. The

---

**Algorithm 1** Multi-resolution training and sampling via preconditioning.

---

**Require:** Boolean FREEZE.
1: **for** $i \leftarrow \{1, \ldots, J\}$ **do**
2:     **if** $i > 1$ **then**
3:         Precondition on $U_{i-1}$.
4:     Train $U_i$
5:     **if** FREEZE is True **then**
6:         Freeze $E_i^\theta$ and $D_i^\theta$ (fix parameters).

---

model was trained on MNIST-Triangular, a custom version of MNIST with the digit and support over a right-angled triangle. While we observe qualitatively correct samples from this dataset, we note that these are obtained with no hyperparameter tuning to improve their fidelity. This experiment has a potentially large scope as it paves the way to designing natural U-Net architectures on tessellations of complicated geometries such as spheres, manifolds, fractals, or CW-complexes.

## 6 Conclusion

We provided a framework for designing and analysing U-Nets. Our work has several limitations: We put particular emphasis on Hilbert spaces as the decoder spaces. We focus on orthogonal wavelet bases, in particular of $L^2(\mathbb{X})$ or $L^2(\triangle)$, while other bases could be explored (e.g. Fourier frequencies, radial basis functions). Our framework is motivated by subspace preconditioning, with requires the user to actively design and choose which subspaces they wish to precondition on. Our analysis of signal and noise concentration in Theorem 2 has been conducted for a particular, yet common choice of denoising diffusion model, with one channel, and with functions supported on one spatial dimension only, but can be straight-forwardly extended with the use of a Kronecker product. Little to no tuning is performed how to allocate the saved parameters in Multi-ResNet in §5.1. We design and empirically demonstrate U-Nets on triangles, while one could choose a multitude of other topological structures. Lastly, future work should investigate optimal choices of $\mathcal{U}$ for domain-specific problems.

## Acknowledgments and Disclosure of Funding

Christopher Williams acknowledges support from the Defence Science and Technology (DST) Group and from a ESPRC DTP Studentship. Fabian Falck acknowledges the receipt of studentship awards from the Health Data Research UK-The Alan Turing Institute Wellcome PhD Programme (Grant Ref: 218529/Z/19/Z), and the Enrichment Scheme of The Alan Turing Institute under the EPSRC Grant EP/N510129/1. Chris Holmes acknowledges support from the Medical Research Council Programme Leaders award MC_UP_A390_1107, The Alan Turing Institute, Health Data Research, U.K., and the U.K. Engineering and Physical Sciences Research Council through the Bayes4Health programme grant. Arnaud Doucet acknowledges support of the UK Defence Science and Technology Laboratory (Dstl) and EPSRC grant EP/R013616/1. This is part of the collaboration between US DOD, UK MOD and UK EPSRC under the Multidisciplinary University Research Initiative. Saifuddin Syed and Arnaud Doucet also acknowledge support from the EPSRC grant EP/R034710/1.

The authors report no competing interests.

This research is supported by research compute from the Baskerville Tier 2 HPC service. Baskerville is funded by the EPSRC and UKRI through the World Class Labs scheme (EP/T022221/1) and the Digital Research Infrastructure programme (EP/W032244/1) and is operated by Advanced Research Computing at the University of Birmingham.

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

# Appendix for *A Unified Framework for U-Net Design and Analysis*

## A  Theoretical Details and Technical Proofs

### A.1  Proofs of theoretical results in the main text

**Theorem 1** Suppose $U_i^*$ and $U^*$ are solutions of the $L^2$ regression problem. Then, $\mathcal{L}_{i|j}(U_i^*)^2 \leq \mathcal{L}_{|j}(U^*)^2$ with equality as $i \to \infty$. Further, if $Q_i U^*$ is $V_i$-measurable, then $U_i^* = Q_i U^*$ minimises $\mathcal{L}_i$.

*Proof.* Let $\mathcal{U} = (\mathcal{V}, \mathcal{W}, \mathcal{E}, \mathcal{D}, \mathcal{P}, U_0)$ be a U-Net in canonical form, that is where each encoder map $E_i$ in $\mathcal{E}$ is set to be the identity map, and further assume $W$ is a Hilbert space with $\mathcal{W}$ being a sequence of subspaces spanned by orthogonal basis vectors for $W$. For a $V$-valued random variable $v$, define the $\sigma$-algebra

$$\mathcal{H}_j \coloneqq \sigma(P_j v) = \sigma(v_j). \tag{4}$$

Now the filtration

$$\mathcal{H}_1 \subset \mathcal{H}_2 \subset \cdots \tag{5}$$

increases to $\mathcal{H} \coloneqq \sigma(v)$. These are the $\sigma$-algebras generated by $v_j$. For data $w \in W$, we may define the losses

$$\mathcal{L}_{i|j}^2(U_{i|j}) \coloneqq \frac{1}{|\mathcal{S}|} \sum_{(w_i, v_j) \in \mathcal{S}} \|w_i - U_{i|j}(v_j)\|^2, \tag{6}$$

for $U_{i|j} : V_j \mapsto W_i$ where $w_i = Q_i w$. Let $\mathcal{F}_{i|j}$ be the set of measurable functions from $V_j$ to $W_i$, then the conditional expectation $\mathbb{E}(w_i|v_j)$ is given by the solution of the regression problem

$$\mathbb{E}(w_i|v_j) = \underset{U_{i|j} \in \mathcal{F}_{i|j}}{\arg\min} \mathcal{L}_{i|j}^2(U_{i|j}), \tag{7}$$

where the conditioning is respect to the $\sigma$-algebra $\mathcal{H}_j$ for the response variable $w_i$. For a fixed $i < i'$ and any $j > 0$ we have that

$$\mathcal{L}_{i'|j}^2(U_{i'|j}) = \frac{1}{|\mathcal{S}|} \sum_{(w_{i'}, v_j) \in \mathcal{S}} \|w_{i'} - U_{i'|j}(v_j)\|^2 \tag{8}$$

$$= \frac{1}{|\mathcal{S}|} \sum_{(w_{i'}, v_j) \in \mathcal{S}} \left( \|Q_i(w_{i'} - U_{i'|j}(v_j))\|^2 + \|Q_i^\perp(w_{i'} - U_{i'|j}(v_j))\|^2 \right) \tag{9}$$

$$= \mathcal{L}_{i|j}^2(Q_i U_{i'|j}) + \frac{1}{|\mathcal{S}|} \sum_{(w_{i'}, v_j) \in \mathcal{S}} \|Q_i^\perp(w_{i'} - U_{i'|j}(v_j))\|^2. \tag{10}$$

Any $U_{i'|j}$ admits the parameterisation $U_{i'|j} = U_{i|j} + U_{i'|j}^{\perp, i}$ where $U_{i|j}$ is in $\mathcal{F}_{i|j}$ and $U_{i'|j}^{\perp, i} \in \mathcal{F}_{i'|j}$ and vanishes on $W_i$. Under this parameterisaiton we gain

$$\mathcal{L}_{i'|j}^2(U_{i'|j}) = \mathcal{L}_{i|j}^2(U_{i|j}) + \frac{1}{|\mathcal{S}|} \sum_{(w_{i'}, v_j) \in \mathcal{S}} \|Q_i^\perp(w_{i'} - U_{i'|j}^{\perp, i}(v_j))\|^2, \tag{11}$$

and further,

$$\underset{U_{i'|j} \in \mathcal{F}_{i'|j}}{\arg\min} \mathcal{L}_{i'|j}^2(U_{i'|j}) = \underset{U_{i|j} \in \mathcal{F}_{i|j}}{\arg\min} \mathcal{L}_{i|j}^2(U_{i,j}) + \underset{U_{i'|j}^{\perp, i}}{\arg\min} \frac{1}{|\mathcal{S}|} \sum_{(w_{i'}, v_j) \in \mathcal{S}} \|Q_i^\perp(w_{i'} - U_{i'|j}^{\perp, i}(v_j))\|^2. \tag{12}$$

In particular, when $i' \to \infty$, if we define $\mathcal{L}_{|j}$ to be the loss on $W$ then

$$\underset{U_{|j} \in \mathcal{F}_{|j}}{\arg\min} \mathcal{L}_{|j}^2(U_{|j}) = \underset{U_{i|j} \in \mathcal{F}_{i|j}}{\arg\min} \mathcal{L}_{i|j}^2(U_{i|j}) + \underset{U_{|j}^{\perp, i}}{\arg\min} \sum_{(w, v_j) \in \mathcal{S}} \|Q_i^\perp(w - U_{|j}^{\perp, i}(v_j))\|^2. \tag{13}$$

So for any $j$, we have that $\mathcal{L}_{i|j}(U^*_{i|j}) \leq \mathcal{L}_{|j}(U^*_{|j})$. Further, as $i \to \infty$, the truncation error

$$\sum_{(w,v_j)\in\mathcal{S}} \|Q_i^\perp(w - U_{|j}^{\perp,i}(v_j))\|^2, \tag{14}$$

tends to zero. Now assume that $Q_iU^*$ is $V_i$-measurable then $Q_iU^* \in \mathcal{F}_{i|j}$ and if this did not minimise $\mathcal{L}_{i|j}$, then choosing the minimiser of $\mathcal{L}_{i|j}$ and constructing $U = U_i^* + U^{\perp,i}$ will have $\mathcal{L}^2(U) < \mathcal{L}^2(U^*)$, contradicting $U^*$ being optimal.

$\square$

**Proposition 1** If $\mathcal{U}$ is a residual U-Net, then $U_i$ is a ResNet preconditioned on $U_i^{\mathrm{pre}}(v_i) = U_{i-1}(\tilde{v}_{i-1})$, where $\tilde{v}_{i-1} = P_{i-1}(E_i(v_i))$.

*Proof.* For a ResNet $R(v_i) = R^{\mathrm{pre}}(v_i) + R^{\mathrm{res}}(v_i)$ select $R^{\mathrm{pre}}(v_i) = U_{i-1}(\tilde{v}_{i-1})$ where $\tilde{v}_{i-1} = P_{i-1}(E_i(v_i))$.

$\square$

**Theorem 2** For time $t \geq 0$ and $j \geq i$, $Q_iX_j(t) \stackrel{d}{=} X_i(t)$. Furthermore if $X_i(t) = \sum_{j=0}^i \widehat{X}^{(j)}(t) \cdot \widehat{\phi}_j$, be the decomposition of $X_i(t)$ in its Haar wavelet frequencies (see Appendix C). Each component $\widehat{X}^{(j)}(t)$ of the vector has variance $2^{j-1}$ relative to the variance of the base Haar wavelet frequency.

*Proof.* Let $X_i \in V_i$ represented in the standard basis $\Phi = \{\phi_k : k = 1, \ldots, 2^i\}$ giving

$$X_i = \sum_k X_i^{(k)}\phi_k. \tag{15}$$

To transform this into its Haar wavelet representation where average-pooling is conjugate to basis projection, we can use the map $T_i : V_i \mapsto V_i$ defined by

$$T_i = \Lambda_i H_i, \tag{16}$$

where $H_i$ is the Haar-matrix on resolution $i$ and $\Lambda_i$ is a diagonal scaling matrix with entries

$$(\Lambda_i)_{k,k} = 2^{-i+j-1}, \quad \text{if } k \in \{2^{j-1}, \ldots, 2^j\}, \tag{17}$$

for $j > 0$ and equal to $2^{-i}$ for the initial case $j = 0$. For example, the matrix for a four-pixel image is

$$2^{-2} \cdot \begin{pmatrix} 1 & 0 & 0 & 0 \\ 0 & 1 & 0 & 0 \\ 0 & 0 & 2^1 & 0 \\ 0 & 0 & 0 & 2^1 \end{pmatrix} \begin{pmatrix} 1 & 1 & 1 & 1 \\ 1 & 1 & -1 & -1 \\ 1 & -1 & 0 & 0 \\ 0 & 0 & 1 & -1 \end{pmatrix} = \Lambda_2 H_2. \tag{18}$$

Now given the vector of coefficients $\mathbf{X}_i = (X_i^{(k)})_{k=1}^{2^i}$, the coefficients in the Haar frequencies can be given by

$$\widehat{\mathbf{X}}_i := T_i(\mathbf{X}_i). \tag{19}$$

We define the vectors

$$(\widehat{X}_j)_k = (T_i(\mathbf{X}_i))_{k+2^{j-1}}, \tag{20}$$

for $j \in \{1, \ldots, 2^{j-1}\}$. Now we may write $X_i$ in terms of its various frequency components as

$$X_i = \sum_{j=0}^i \widehat{X}^{(j)} \cdot \widehat{\phi}_j. \tag{21}$$

Suppose that $X_i$ is a time varying random variable on $V_i$ that evolve according to

$$X_i^{(k)}(t) := \sqrt{1 - \alpha_t}X_i^{(k)} + \sqrt{\alpha_t}\varepsilon^{(k)}, \tag{22}$$

when represented in its pixel basis where $\varepsilon^{(k)}$ is a standard normally distributed random variable. We may represent $X_i$ in its frequencies, with a random and time dependence on the coefficient vectors,

$$X_i = \sum_{j=0}^{i} \widehat{X}^{(j)}(t) \cdot \widehat{\phi}_j. \tag{23}$$

To analyse the variance of $\widehat{X}^{(j)}(t)$, we may analyse the variance of a standard normally distributed random vector $\varepsilon_i = (\varepsilon^k)_{k=1}^{2^i}$ under the mapping $T_i$. Due to the symmetries in the matrix representation for $T_i$, the variance for each element of $\widehat{X}^{(j)}(t)$ is the same, and by direct computation

$$\mathrm{VAR}\,(T_i\varepsilon_i)_k = 2^{-i+j-1} \quad \text{for } i > 0, \ k \in \{2^{j-1}, \dots, 2^j\}, \tag{24}$$

and $2^{-i}$ for when $i = 0$. To see this note that

$$\mathrm{VAR}\,(T_i\varepsilon_i) = \mathrm{VAR}(\Lambda_i H_i \varepsilon_i) = \Lambda_i^2 \mathrm{VAR}(H_i \varepsilon_i), \tag{25}$$

as $\Lambda_i$ is a diagonal matrix. Now each of the elements of $\varepsilon_i$ are independent and normally distributed, so if $(H_i)_{k,:}$ is the $k^{th}$ row of $H_i$, then

$$\mathrm{VAR}(H_i\varepsilon_i)_k = \mathrm{VAR}((H_i)_{k,:}\varepsilon_i) = \|(H_i)_{k,:}\|_0, \tag{26}$$

where $\|(H_i)_{k,:}\|_0$ is the amount of non-zero elements of $(H_i)_{k,:}$ as each entry of $H_i$ is 0, or $\pm 1$. For $(H_i)_{k,:}$ where $k \in \{2^{j-1}, \dots, 2^j\}$ there are $2^{i-j+1}$ entries. Further, in this position we have $(\Lambda_i)_k = 2^{-i+j-1}$, so putting this together finishes the count. Now see that for the evolution of $X_i^{(k)}(t)$, the random part of Eq. (22) is simply a Gaussian random vector multiplied by $\sqrt{\alpha_t}$. Therefore we have that $\mathrm{VAR} X_i^{(k)}(t) = \alpha_t 2^{-i+j-1}$. To get the relative frequency, for $k$ we divide this variance by the variance of the base frequency, the zeroth resolution. This yields $\alpha_t 2^{-i+j-1}/\alpha_t 2^{-i} = 2^{j-1}$.

Now we show how the diffusion on a given resolution $i$ can be embedded into a higher resolution of $i + 1$, which maintains consistency with the projection map $Q_i$. Define the diffusion on $i + 1$ to be

$$X_{i+1}^{(k)}(t) := \sqrt{1 - \alpha_t} X_{i+1}^{(k)} + \sqrt{2\alpha_t} \varepsilon^{(k)} \tag{27}$$

Let $\widehat{X}_i^{(j)}(t)$ be the Haar coefficients for the inital diffusion defined on resolution $i$ and $\widehat{X}_i^{(j)}(t)$ be the coefficients defined on resolution $i + 1$. Both of these random vectors are linear transformations of Gaussian random vectors, so we need only confirm that the means and variances on the first $j$ entries agree to show that these are equal in distribution. If $X_i^{(k)} = \frac{X_{i+1}^{(k)} + X_{i+1}^{(k+1)}}{2}$, that is the initial data on resolution $i + 1$ averages to the initial data on resolution $i$, then by construction the means of the random vectors $\widehat{X}_i^{(j)}(t)$ and $\widehat{X}_{i+1}^{(j)}(t)$ agree for the first $i$ resolutions. Note that the variances for the components of the noise for either process are $2^{-i+j-1}$ when the diffusion on $i + 1$ is given by Eq. (27). In general, for the process on $i + \ell$ to embed into our original diffusion on resolution $i$ we require scaling the noise term by a factor of $2^{\ell/2}$. Again, this shows that the noising process on a frequency $i$ has the natural extension noising the high-frequency components exponentially faster than the reference frequency $i$. $\qquad\square$

We may comment on how this is represented in its natural sequence space. Assume that $X_i \in V_i \subset L^2([0, 1])$. There is a natural mapping from $V_i$ to $\ell_2$ defined by

$$\ell_2 := \{s| \sum_{k=0}^{\infty} s_k^2 < \infty\} \tag{28}$$

through the mapping $\pi : L^2([0, 1]) \mapsto \ell_2$ via

$$\pi(X_i) = (\widehat{X}^{(1)}, \widehat{X}^{(2)}, \dots). \tag{29}$$

For elements $X_i \in V_i$ the image of $\pi$ is not surjective and has image contained in $\ell_{00}$ — the set of infinite sequences which eventually are zero. In this case we have a finite dimensional mapping. Theorem 2 simply states that the forward noising process in a diffusion model, under transformation into its Haar basis, has variance

$$\mathrm{VAR}\,(\pi X_i)_k = 2^{-i+j-1} \quad \text{for } i > 0, \ k \in \{2^{j-1}, \dots, 2^j\}, \tag{30}$$

implying that the sequence image of the datum has monotonically increasing variance in the Haar wavelet sequence space.

## A.2 Diffusions on the sphere.

Suppose that we have $L^2(\mathbb{X})$ valued data on the globe and we would like to form a diffusion model on this domain. Instead of constructing a local change of coordinate system over a manifold and projecting our diffusion to a square domain[16], we could simply design our U-Net to encode our geometry. As an example of this, we can take a standard triangulation of the globe, then apply our triangular basis U-Net directly to our data.

For the data on the globe, we could then choose a resolution $i$ to refine our triangular domain to, see Figure 9 for a refinement level of $i = 2$ which could be used to tessellate the globe. For each 'triangle pixel' in our refinement, we could define the diffusion process

$$X_t^{\mathbf{i}} = \sqrt{1 - \widetilde{\alpha}} X_0^{\mathbf{i}} + \sqrt{\widetilde{\alpha}} \varepsilon^{\mathbf{i}}, \tag{31}$$

where $\mathbf{i}$ is a string of length $i + 1$ encoding which triangle on the globe the pixel is in, then which 'triangular pixel' we are in. We can construct Haar wavelets on the triangle (see Figure 3.3), and hence create natural projection maps in $L^2$ over this domain. In this case $W_0$ is the span of constant functions over each large triangle for the tessellation of the globe, and each $W_i$ is $W_0$ along with the span of the wavelets of resolution $i$ over each triangle. Like in the standard diffusion case, we choose $V_i = W_i$ and can use the mapping given in Section 5.2 to construct the natural U-Net design for this geometry, and for this basis. Similarly, for any space that we can form a triangulation for, we can use this construction to create the natural Haar wavelet space for this geometry, define a diffusion analogously, and implement the natural U-Net design for learning.

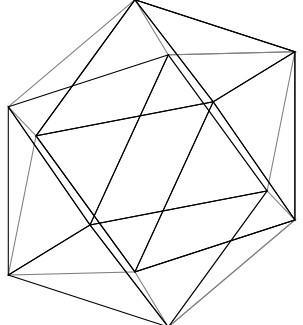 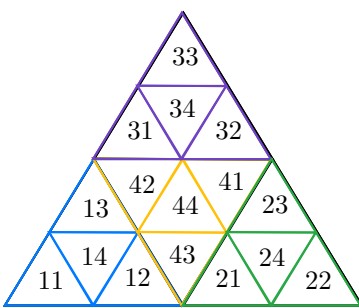

Figure 9: Triangulation of the globe with a self-similar triangle which admits a Haar wavelet basis. Upon refinement through the self-similarity of the triangle we receive finer and finer approximations of the globe, and of functions over it.

## A.3 U-Nets on Finite Elements

PDE surrogate modelling is a nascent research direction in machine learning research. We have seen in our PDE experiment (see Section 5) that our Multi-ResNet design outperforms Residual U-Nets on a square domain. We would like to mimic this in more general geometries and function spaces. Here, we will show on the unit interval how to design a U-Net over basis functions which enforce boundary constraints and smoothness assumptions, mimicking the design of Finite Elements used for PDE solvers.

Recall our model problem

$$\Delta u = f, \qquad\qquad u(0) = u(1) = 0, \tag{32}$$

which when multiplied by a test function $\phi \in \mathcal{H}_0^1([0, 1])$ and integrated over the domain puts the equation into its weak form, and by integration by parts

$$-\langle \phi', u' \rangle_{\mathcal{H}_0^1} = \int \phi f, \tag{33}$$

we recover the standard bilinear representation of Laplaces equation. Now if our solution $u \in \mathcal{H}_0^1([0, 1])$ and we have an orthogonal basis $\{\phi_{i,k}\}$ for $\mathcal{H}_0^1([0, 1])$, then the coefficients for the basis elements of the solution can be solved by finding the solution of a linear system with a diagonal

mass matrix. Thus, in this way, the information needed to solve the linear system up to some finite resolution $i$, only depends on data up to that resolution in this weak form. This is a Galerkin truncation of the PDE used for forward solvers on finite resolutions, but also gives us the ideal assumptions we need for the construction of our U-Net (the measurability constraint in Theorem 1).

For an explicit example, we may make an orthogonal basis for $\mathcal{H}_0^1([0,1])$ with the initial basis function [Left] of Figure 4 and its refined children on the [Right] of Figure 4.

In general we can refine further and create a basis of resolution $i$ through

$$\phi_{k,j}(x) = \phi(2^k x + j/2^k), \qquad \phi(x) = 2x \cdot 1_{[0,1/2)}(x) + (2 - 2x) \cdot 1_{[1/2,1]}(x),$$

which are precisely the integrals of the Haar wavelets up to a given resolution. We would again construct $W_i$ in our U-Net as the span of the first resolution $i$ basis functions, where we now have additionally encoded the boundary constraints and smoothness requirements of our PDE solution into our U-Net design.

# B  Additional experimental details and results

In this section, we provide further details on our experiments, and additional experimental results.

**Model and training details.**  We used slightly varying U-Net architectures on the different tasks. While we refer to our code base for details, they generally feature a single residual block per resolution for $E_i$ and $D_i$, with 2 convolutional layers, and group normalisation. We in general choose $P_i$ as average pooling. We further use loss functions and other architectural components which are standard for the respective tasks. This outlined U-Net architecture often required us to make several changes to the original repositories which we used. This may also explain small differences to the results that these repositories reported; however, our results are comparable. When using Algorithm 1 during staged training, we require one 'head' and 'tail' network on each resolution which processes the input and output respectively, and hence increases the number of parameters of the overall model relative to single-stage training.

**Hyperparameters and hyperparameter tuning.**  We performed little to no hyperparameter tuning in our experiments. In particular, we did not perform a search (e.g. grid search) over hyperparameters. In general, we used the hyperparameters of the original repositories as stated in Appendix D, and changed them only when necessary, for instance to adjust the number of parameters so to enable a fair comparison. There is hence a lot of potential to improve the performance of our experiments, for instance of the Multi-ResNet, or the U-Net on triangular data. We refer to our code repository for specific hyperparameter choices and further details, in particular the respective `hyperparam.py` files.

**Evaluation and metrics.**  We use the following three performance metrics in our experimental evaluation: FID score [44], rollout mean-squared-error (r-MSE) [6], and Sørensen–Dice coefficient (Dice) [45, 46]. As these are standard metrics, we only highlight key points that are worth noting. We compute the FID score on a holdout dataset not used during training, and using an evaluation model where weights are updated with the training weights using an exponential moving average (as is common practice). The r-MSE is computed as an MSE over pieces of the PDE trajectory against its ground-truth. Each piece is predicted in an autoregressive fashion, where the model receives the previous predicted piece and historic observations as input [6]. All evaluation metrics are in general computed on the test set and averaged over three random seeds after the same number of iterations in each table, if not stated otherwise. The results are furthermore not depending on these reported evaluation metrics: in our code base, we compute and log a large range of other evaluation metrics, and we typically observe similar trends as in those reported.

### B.1 Analysis 1: The role of the encoder in a U-Net

In this section, we provide further experimental results analysing the role of the encoder, and hence Residual U-Nets with Multi-ResNets. We realise the Multi-ResNet by replacing the parameterised encoder with a multi-level Discrete Wavelet Transform (DWT). More specifically, the skip connection on resolution $j$ relative to the input resolution is computed by taking the lower-lower part of the DWT at level $j$ ($LL_j$), and then inverting $LL_j$ to receive a coarse, projected version of the original input image.

**Generative modelling with diffusion models.** In Table 2, we analyse the role of the encoder in generative modelling with diffusion models. Following our analysis on PDE modelling and image segmentation, we compare (Haar wavelet) Residual U-Nets with (Haar wavelet) Multi-ResNets where we add the saved parameters in the encoder back into the decoder. We report performance in terms of an exponential moving average FID score [55] on the test set. In contrast to the other two tasks, we find that diffusion models benefit from a parameterised encoder. In particular, the Multi-ResNet does not outperform the Residual U-Net with approximately the same number of parameters, as is the case in the other two tasks. Referring to our theoretical results in §3.1, the input space of Haar wavelets may be a suboptimal choice for diffusion models, requiring an encoder learning a suitable change of basis. A second possibility is the FID score itself, which is a useful, yet flawed metric to quantify the fidelity and diversity of images [56, 57]. To underline this point, in Figure 10 we compare samples from two runs with the Residual U-Net and Multi-ResNet as reported in Table 2. We observe that it is very difficult to tell "by eye" which model produces higher quality samples, in spite of the differences in FID. A third possibility is that the allocation of the saved parameters is suboptimal. We demonstrate the importance of beneficially allocating the saved parameters in the decoder in the context of PDE modelling below.

Table 2: Quantitative performance of the (Haar wavelet) Multi-ResNet compared to a classical (Haar wavelet) Residual U-Net in generative modelling with diffusion models on `CIFAR10`. We report FID on the test set.

| Dataset | Neural architecture | # Params. | FID ↓ |
|---|---|---|---|
| **CIFAR10** 32 × 32 | Residual U-Net | 35.5 M | $7.86 \pm 0.25$ |
| | Multi-ResNet, no params. added in dec. (*ours*) | 25.8 M | $14.87 \pm 0.50$ |
| | Multi-ResNet, saved params. added in dec. (*ours*) | 32.4 M | $12.44 \pm 0.22$ |

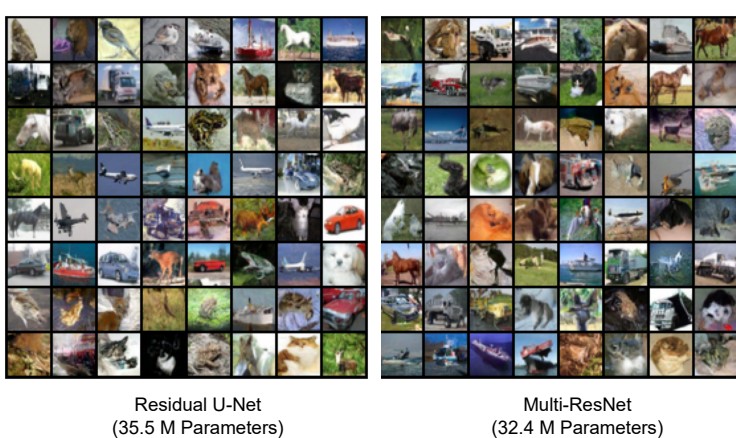

Residual U-Net
(35.5 M Parameters)

Multi-ResNet
(32.4 M Parameters)

Figure 10: Samples of a DDPM-type diffusion model with a Residual U-Net [Left] and a Multi-ResNet [Right]. We trained both models for 1.2 M iterations at which we obtained the samples.

Lastly, we present Figure 7 from the main text at a larger resolution in Figure 11.

## CIFAR10

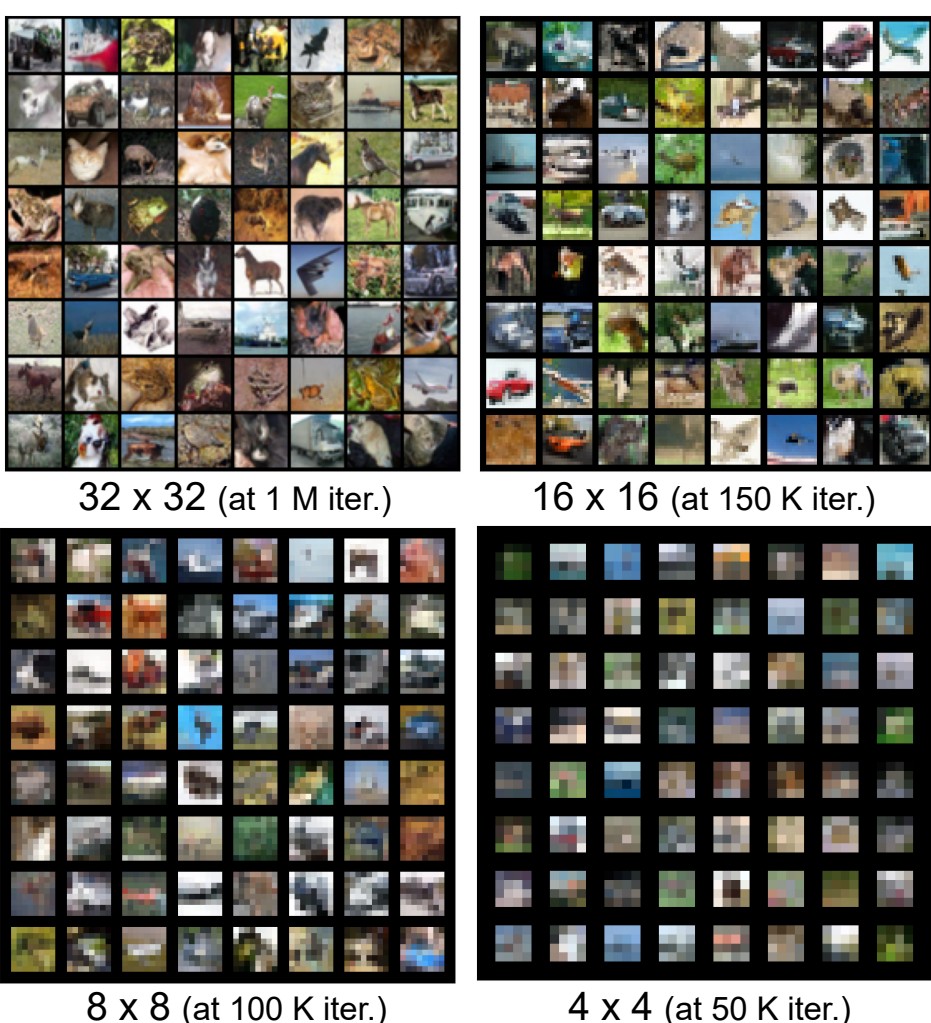

Figure 11: Preconditioning enables multi-resolution training and sampling of diffusion models. We train a diffusion model [8] with a Residual U-Net architecture using Algorithm 1 with four training stages and no freezing on CIFAR10. This Figure is identical with Figure 7, but larger.

**PDE modelling.** We further present additional quantitative results comparing the (Haar wavelet) Multi-ResNet with (Haar wavelet) Residual U-Nets on our PDE modelling tasks. These shall highlight the importance of allocating the saved parameters in Multi-ResNets to achieve competitive performance. In Table 3, we compare two versions of allocating the parameters we save in Multi-ResNets, which have no parameters in their encoder, on the Navier-stokes and Shallow water PDE modelling datasets, respectively. In 'v1', also presented in the main text in Table 1 but repeated for ease of comparison, we add the saved parameters by increasing the number of residual blocks per resolution. This in effect corresponds to more discretisation steps of the underlying ODE on each resolution [20]. In 'v2', we add the saved parameters by increasing the number of hidden channels in the ResNet.

Our results show that 'v1' performs significantly better than 'v2'. In particular, 'v2' does not outperform the Residual U-Net on the two PDE modelling tasks. This indicates that the design choice of how to allocate parameters in $\mathcal{D}$ matters for the performance of Multi-ResNets. We note that in our experiments, beyond this comparison, we did not explore optimal ways of allocating parameters in Multi-ResNets. Hence, future work should explore this aspect thoroughly with large-scale experiments. Due to its 'uni-directional', 'asymmetric' structure, the Multi-ResNet may further

be combined with a direction of transformer-based architectures which aim at outperforming and replacing the U-Net [53, 54]. With reference to our earlier presented, inferior results with diffusion models, it is possible that merely designing a better decoder would make Multi-ResNets superior to classical Residual U-Nets for diffusion models.

Table 3: Quantitative performance of the (Haar wavelet) Multi-ResNet compared to a classical (Haar wavelet) Residual U-Net on two PDE modelling. This table is an augmented version of Table 1 in the main text, where 'v1' indicates the run from the main text, a Multi-ResNet with saved parameters added in the encoder, and 'v2' indicates an alternative parameter allocation. We report Mean-Squared-Error over a rolled out trajectory in time (r-MSE) on the test set, rounded to four decimal digits.

| Dataset | Architecture | # Params. | r-MSE $\downarrow$ |
|---|---|---|---|
| **Navier-stokes** $128 \times 128$ | Residual U-Net | 34.5 M | $0.0057 \pm 2 \cdot 10^{-5}$ |
| | Multi-ResNet, no params. added in dec. (*ours*) | 15.7 M | $0.0107 \pm 9 \cdot 10^{-5}$ |
| | Multi-ResNet, saved params. added in dec. v1 (*ours*) | 34.5 M | $\mathbf{0.0040 \pm 2 \cdot 10^{-5}}$ |
| | Multi-ResNet, saved params. added in dec. v2 (*ours*) | 34.6 M | $0.0093 \pm 5 \cdot 10^{-5}$ |
| **Shallow water** $96 \times 192$ | Residual U-Net | 34.5 M | $0.1712 \pm 0.0005$ |
| | Multi-ResNet, no params. added in dec. (*ours*) | 15.7 M | $0.4899 \pm 0.0156$ |
| | Multi-ResNet, saved params. added in dec. v1 (*ours*) | 34.5 M | $\mathbf{0.1493 \pm 0.0070}$ |
| | Multi-ResNet, saved params. added in dec. v2 (*ours*) | 34.6 M | $0.3811 \pm 0.0091$ |

In Figure 12 we illustrate the full prediction of our Haar wavelet Multi-ResNet when modelling a PDE trajectory simulated from the `Navier-stokes` and `Shallow water` equations, respectively, corresponding to its truncated version in Figure 6 [Left]. We unroll the trajectories over five timesteps for which we predict the current state. Note that we train the Multi-ResNet by predicting the next time step in the trajectory only. We do not condition on previous timesteps.

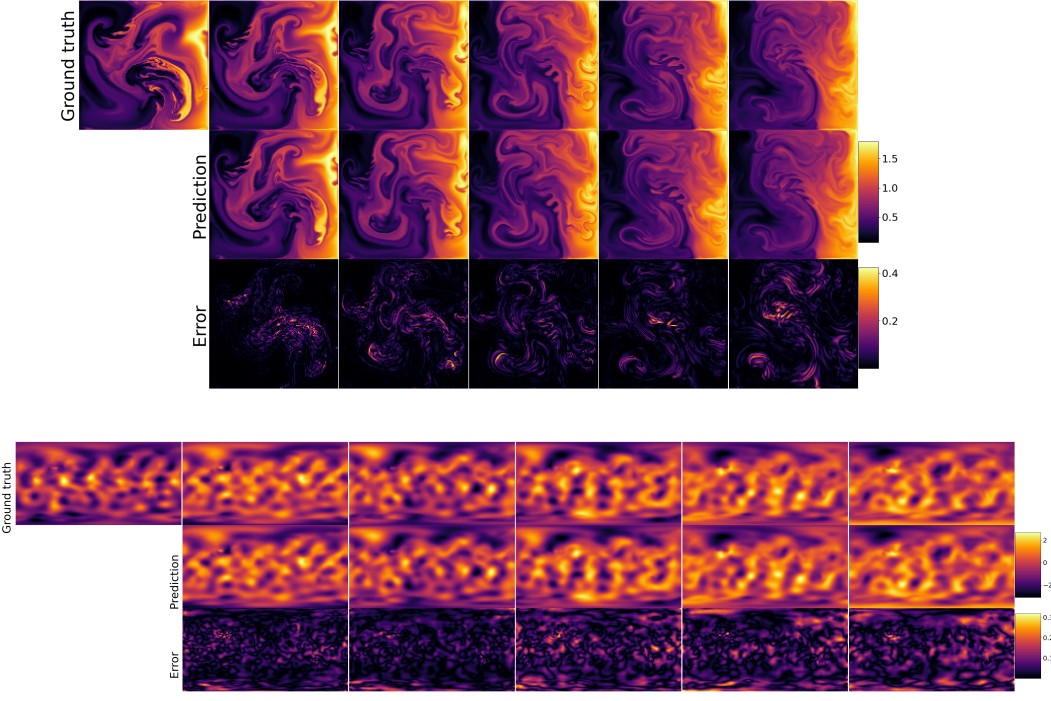

Figure 12: PDE modelling and image segmentation with a Wavelet-encoder Multi-ResNet. Rolled out PDE trajectories (ground-truth, prediction, $L^2$-error between ground-truth and prediction) from the `Navier-Stokes` [top], and the `Shallow-Water` equation [bottom]. This is the complete version of the truncated Figure 6 [Left] showing further timesteps for the trajectory. Figure and code as modified from [6, Figure 1].

We also present results comparing our Multi-ResNet, Residual U-Nets and the Fourier Neural Operator (FNO) [52], another competitive model for PDE modelling, in Table 4. We compare models of similar size in terms of parameter count. We also present the same comparison in PDEArena [6] in Table 5 for reference, noting that our experimental configuration slightly differs. Both tables indicate that U-Nets outperform FNO.

Table 4: Quantitative performance of an FNO model compared to U-Nets. We compare the FNO model to a (Haar wavelet) Multi-ResNets and classical (Haar wavelet) Residual U-Nets with similar number of parameters, on two PDE modelling tasks.

| Dataset | Neural architecture | # Params. | r-MSE $\downarrow$ |
|---|---|---|---|
| **Navier-stokes** $128 \times 128$ | Residual U-Net | 34.5 M | $0.0057 \pm 2 \cdot 10^{-5}$ |
| | Multi-ResNet, saved params. added in dec. | 34.5 M | $\mathbf{0.0040 \pm 2 \cdot 10^{-5}}$ |
| | FNO 128-8 mode8 (*new*) | 33.7 M | $0.0253 \pm 0.0$ |
| **Shallow water** $96 \times 192$ | Residual U-Net | 34.5 M | $0.1712 \pm 0.0005$ |
| | Multi-ResNet, saved params. added in dec. | 34.5 M | $\mathbf{0.1493 \pm 0.0070}$ |
| | FNO 128-8 mode8 (*new*) | 33.7 M | $1.2333 \pm 0.0115$ |

Table 5: Experimental results as reported in PDEArena [6]): Quantitative performance of an FNO model, in comparison to U-Nets of similar size. Values as reported in Table 8 in [7] (5200 trajectories) for Navier-Stokes, and in [Table 2 in [7] (5600 trajectories)] for Shallow water. Here, U-Net 2015 64 clearly outperforms FNO 128-8 mode8. This result is consistent across different numbers of trajectories (dataset sizes), and different data configurations (e.g. velocity function formulation vs. vorticity stream function formulation on Shallow water) in the several tables reported in [7].

| Dataset | Neural architecture | # Params. | r-MSE $\downarrow$ |
|---|---|---|---|
| **Navier-stokes** $128 \times 128$ | U-Net 2015 64 | 31 M | $0.01386 \pm 0.00004$ |
| | FNO 128-8 mode8 | 33.7 M | $0.03836 \pm 0.00037$ |
| **Shallow water** $96 \times 192$ | U-Net 2015 64 | 31 M | $0.1026 \pm 0.0161$ |
| | FNO 128-8 mode8 | 33.7 M | $0.8549 \pm 0.0124$ |

**Image segmentation.** In Figure 13, we present further MRI images with overlayed ground-truth (green) and prediction (blue) masks, augmenting Figure 6 [Right] in the main text. The predictions are obtained from our best-performing Multi-ResNet. Note that some MRI images do not contain any ground-truth lesions.

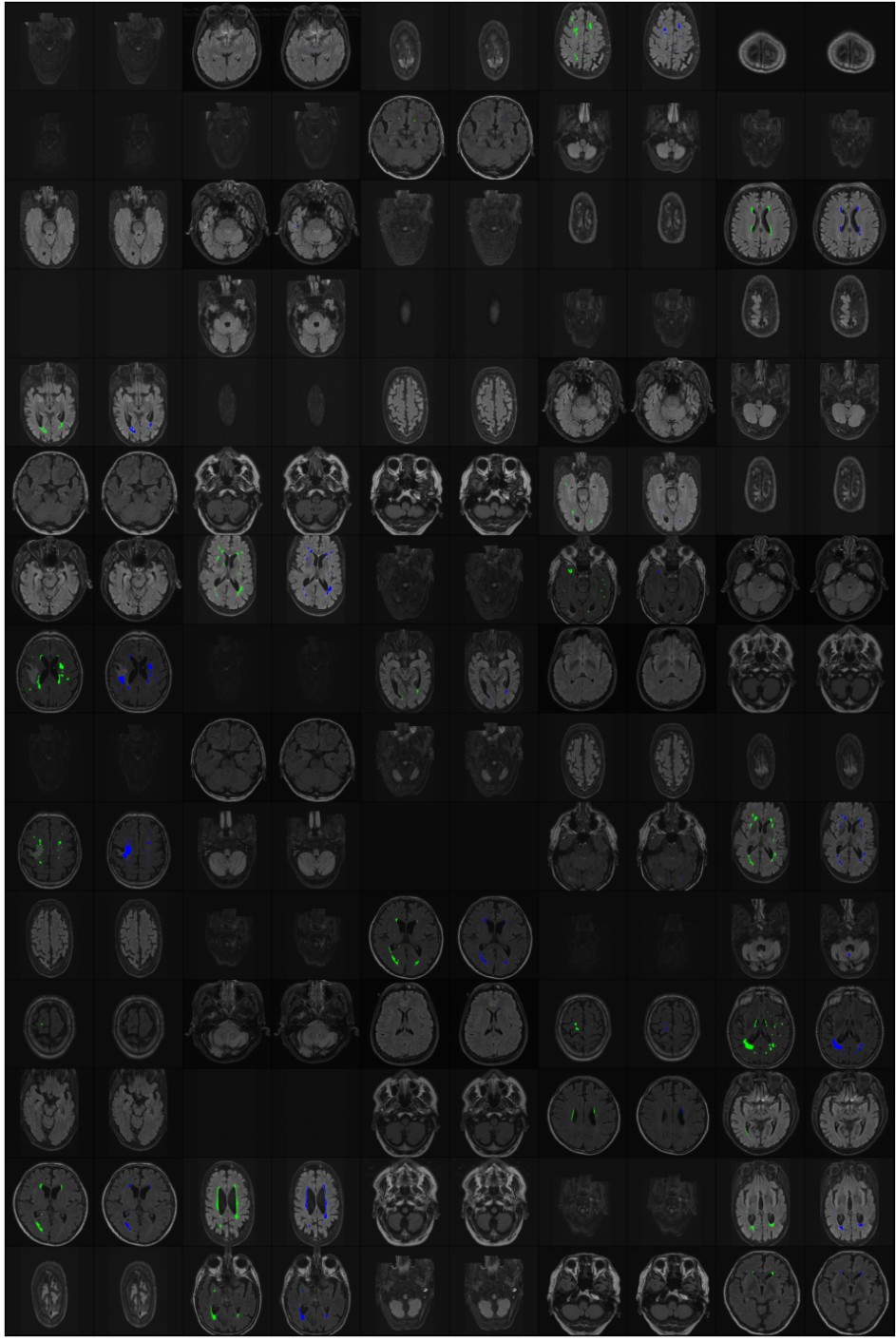

Figure 13: MRI images from WMH with overlayed ground-truth masks (green) and predictions (blue) obtained from our best-performing Multi-ResNet model.

## B.2 Analysis 2: Staged training enables multi-resolution training and inference

We begin by providing further experimental details. For each dataset, we train each resolution for the following number of iterations/epochs: `MNIST` $[5K, 5K, 5K, 5K]$ iterations, `CIFAR10` $[50K, 50K, 50K, 450K]$ iterations, `Navier-Stokes` $[5, 5, 5, 35]$ epochs, `Shallow water` $[2, 2, 2, 14]$ epochs.

**Generative modelling with diffusion models.** In Figs. 14 and 15, we illustrate samples of a diffusion model (DDPM) trained on `MNIST` with Algorithm 1, with and without freezing, respectively. These samples show that in both cases, the model produces reasonable samples. Algorithm 1 with freezing has the advantage that the final model can produce samples on all resolutions, instead of only the intermediate models at the end of each training stage. Preliminary tests showed that as perhaps expected, freezing lower-resolution U-Net weights tends to produce worse performance in terms of quantitative metrics measured on the highest resolution. In particular, in Figure 16 we compare samples and FID scores from a DDPM model trained on `CIFAR10`, with and without freezing. As can be clearly seen from both the FID score and the quality of the samples, training with Algorithm 1 but without the freezing option produces significantly better samples. This is why we did not further explore the freezing option.

MNIST

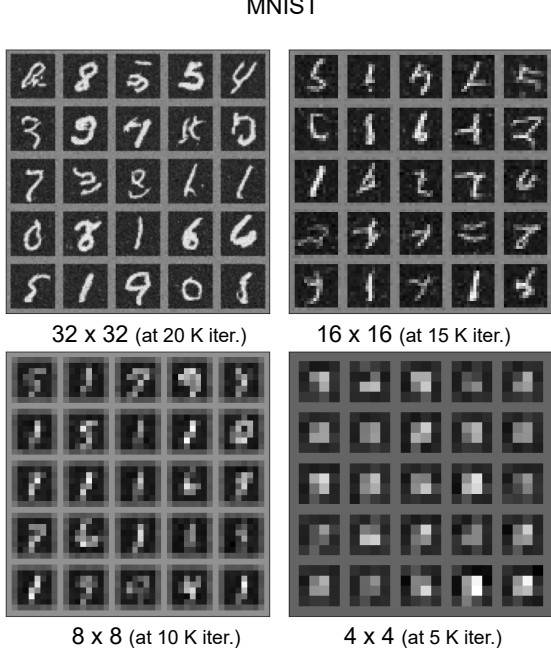

Figure 14: Staged training of Multi-ResNets enables multi-resolution training and sampling of diffusion models. We train a diffusion model [8] with a Residual U-Net architecture using Algorithm 1 with four training stages and *with freezing* on `MNIST` corresponding to Figure 7. We show samples at the end of each training stage.

**PDE modelling.** In Table 6, we investigate Residual U-Nets and Multi-ResNets using Algorithm 1 on `Navier-Stokes` and `Shallow water`. We here find that staged training makes the runs perform substantially worse. In particular, the standard deviation is higher, and some runs are outliers with particularly poor performance. This is in contrast to our experiments on generative modelling with diffusion models where staged training had only an insignificant impact on the resulting performance. We note that we have not further investigated this question, and believe that there are a multitude of ways which could be attempted to stabilise and improve the performance of staged training, for instance in PDE modelling, which we leave for future work.

MNIST

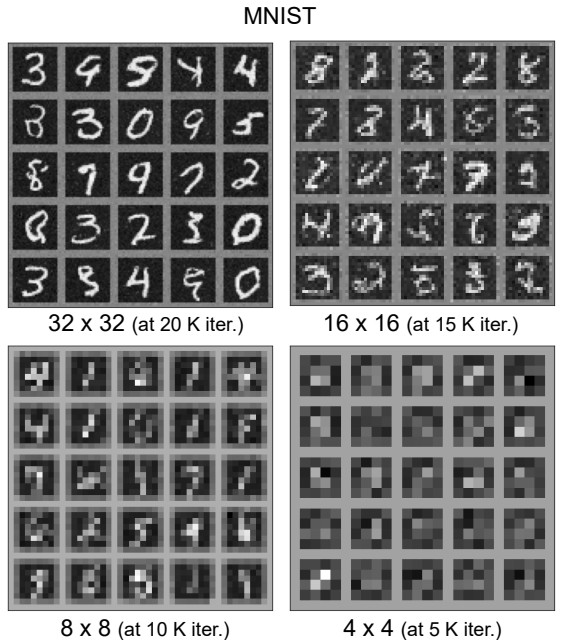

Figure 15: Staged training of Multi-ResNets enables multi-resolution training and sampling of diffusion models. We train a diffusion model [8] with a Residual U-Net architecture using Algorithm 1 with four training stages and *no freezing* on `MNIST` corresponding to Figure 7. We show samples at the end of each training stage.

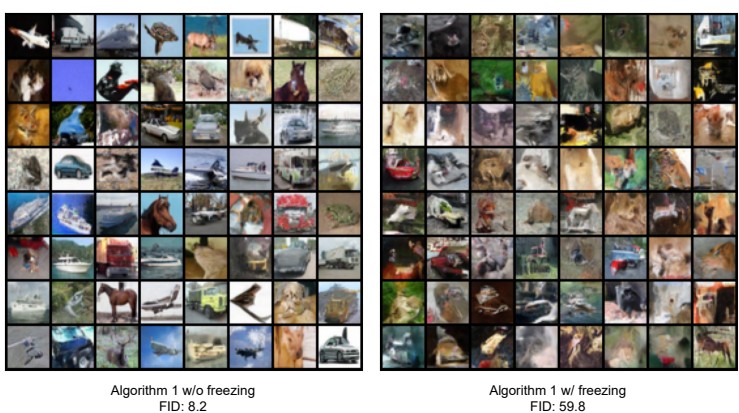

Figure 16: A comparison of samples of a DDPM model [8] with a Residual U-Net trained with Algorithm 1 with [Right] and without [Left] freezing.

Table 6: Staged training with Algorithm 1 on `Navier-Stokes` and `Shallow water`. Quantitative performance of the (Haar wavelet) Multi-ResNet compared to a classical (Haar wavelet) Residual U-Net.

| Dataset | Neural architecture | # Params. | r-MSE ↓ |
|---|---|---|---|
| **Navier-stokes** $128 \times 128$ | Residual U-Net | 37.8 M | $0.0083 \pm 0.0034$ |
| | Multi-ResNet, saved params. added in dec. (*ours*) | 37.8 M | $0.0105 \pm 0.0102$ |
| **Shallow water** $96 \times 192$ | Residual U-Net | 37.7 M | $0.4640 \pm 0.4545$ |
| | Multi-ResNet, saved params. added in dec. (*ours*) | 37.7 M | $0.7715 \pm 0.8327$ |

### B.3 Analysis 3: U-Nets encoding topological structure

**Experimental details.** We begin by discussing `MNIST-Triangular`, the dataset we used in this experiment. In a nutshell, `MNIST-Triangular` is constructed by shifting the MNIST $28 \times 28$ dimensional squared digit into the lower-left half of a $64 \times 64$ squared image with similar background colour as in MNIST in that lower-left half of triangular shape. In the upper-right half, we use a gray background colour to indicate that the image is supported only on the lower-left of the squared image. We illustrate `MNIST-Triangular` in Figure 17. We note that `MNIST-Triangular` has more than four times the number of pixels compared to `MNIST`, yet we trained our DDPM U-Net with hyperparameters and architecture for `MNIST` without hyperparameter tuning, explaining their perhaps slightly inferior sample quality in Figure 8.

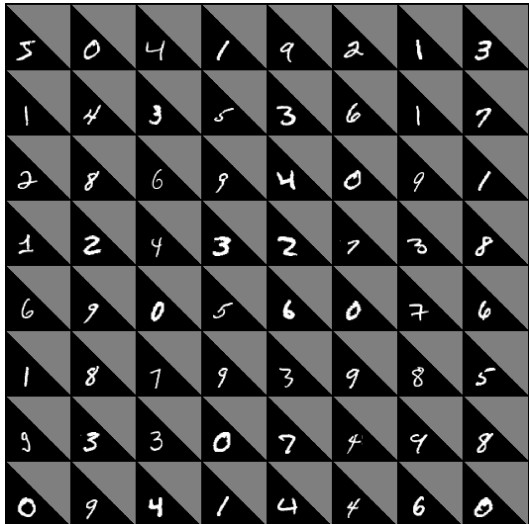

Figure 17: Example images from the `MNIST-Triangular` dataset.

To process the data into the subspaces $\mathcal{W} = (W_i)$ given in Section 3.3, we need to formulate the 'triangular pixels' shown in Figure 3.3. To do this we use the self-similarity property of the triangular domain to divide the data into a desired resolution. For each image, we then sample the function value at the center of our triangular pixel and store this in a lexicographical ordering corresponding to the *codespace address* [58] (e.g. '21') of that pixel. In Figure 18 we illustrate the coding map at depth two.

Once we have made this encoding map, we are able to map our function values into an array and perform the projections on $V = W$ by push-forwarding through projection of the Haar wavelets on the triangle to our array through the lexicographic ordering used. For instance, we show in Figure 19

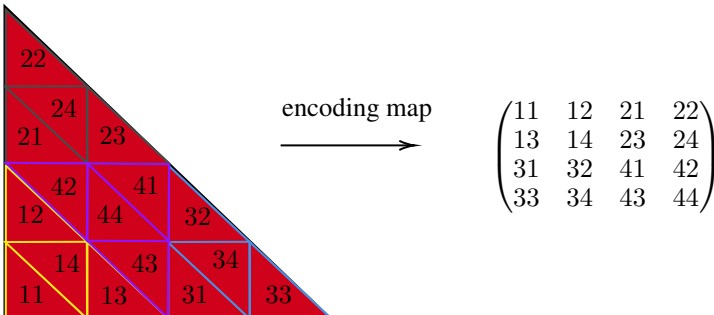

Figure 18: The coding map from the triangular Haar wavelets to their code-space addresses. Such a construction can always be made on a self-similar object with certain separation properties, such as the *Open Set Condition* [59].

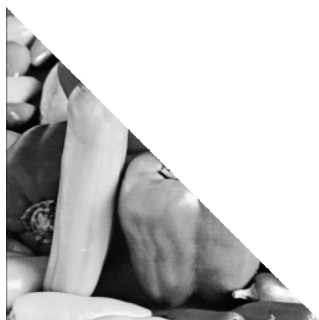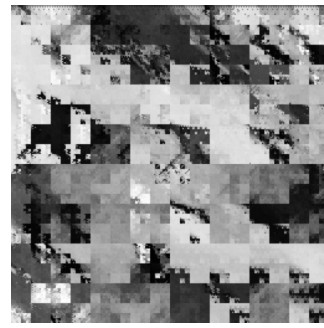

Figure 19: An example of the encoding map taking data from a triangular domain and mapping it to an array of each 'triangular pixel' value under lexicographical ordering.

the array, plotted as an image, revealing that this processing is inherently different to diffusions on a square domain.

This transformation, when defined on the infinite resolution object, is discontinuous on a dense set of $\triangle$, yet encodes all of the necessary data to perform a diffusion model over in our triangular MNIST model seen in Figure 8. This is because the transformation we are using is still continuous almost everywhere, and so the parts of the signal that we are losing accounts for a negligible amount of the function approximation in the diffusion process.

## B.4 Ablation studies

In several ablation studies, we further analysed Multi-ResNets and our experimental results. We present these below.

### B.4.1 Ablation 1: The importance of skip connections in U-Nets

We begin by presenting a key result. As Multi-ResNets perform a linear transformation in the encoder, which significantly reduces its expressivity, one might hypothesise that the skip connections in a U-Net can be removed entirely. Our results show that skip connections are crucial in U-Nets, particularly in Multi-ResNets.

In Table 7 we compare Residual U-Nets and Multi-ResNets with and without skip connections focussing on PDE Modelling (`Navier-stokes` and `Shallow water`). The 'without skip connections' case is practically realised by feeding zero tensors along the skip connections. This has the benefit of requiring no change to the architecture enabling a fair comparison, while feeding no information via the skip connections as if they had been removed. We find that performance significantly deteriorates when using no skip connections. This effect is particularly strong in Multi-ResNets. Multi-ResNets cannot compensate the lack of skip connections by learning the encoder in such a way that representations learned on the lowest resolution space $V_0$ account for not having access to higher-frequency information via the skip connections when approximating the output via $D_i$ in the output space $W_i$ on a higher resolution space. This result shows that the encoder actually compresses the information which it provides to the decoder. The decoder crucially depends on this compressed input.

Table 7: Skip connections in U-Nets are crucial. We compare two Multi-ResNets, (Haar wavelet) Residual U-Nets and (Haar wavelet) Multi-ResNets, with and without skip connections, focussing on the two PDE modelling datasets (`Navier-Stokes` and `Shallow water`).

| | Neural architecture | # Params. | r-MSE $\downarrow$ |
|---|---|---|---|
| **Navier-stokes** $128 \times 128$ | Residual U-Net w/ skip con. | 34.5 M | $\mathbf{0.0057 \pm 2 \cdot 10^{-5}}$ |
| | Residual U-Net w/o skip con. | 34.5 M | $0.0078 \pm \cdot 10^{-5}$ |
| | Multi-ResNet w/ skip con. | 34.5 M | $\mathbf{0.0040 \pm 2 \cdot 10^{-5}}$ |
| | Multi-ResNet w/o skip con. | 34.5 M | $0.0831 \pm 0.1080$ |
| **Shallow water** $128 \times 128$ | Residual U-Net w/ skip con. | 34.5 M | $\mathbf{0.1712 \pm 0.0005}$ |
| | Residual U-Net w/o skip con. | 34.5 M | $0.4950 \pm 0.02384$ |
| | Multi-ResNet w/ skip con. | 34.5 M | $\mathbf{0.1493 \pm 0.0070}$ |
| | Multi-ResNet w/o skip con. | 34.5 M | $0.6302 \pm 0.01025$ |

### B.4.2 Ablation 2: The importance of preconditioning in U-Nets

In this section, we are interested in analysing the importance of preconditioning across subspaces in U-Nets. To this end, we analyse whether and how much a U-Net benefits from the dependency in the form of preconditioning between its input and output spaces $\mathcal{V}$ and $\mathcal{W}$. In Table 8, we compare a Residual U-Net with multiple subspaces with a ResNet with only one subspace on `Navier-Stokes` and `Shallow water`, respectively. We obtain the ResNet on a single subspace from the Residual U-Net by replacing $P_i$ as well as the (implicit) embedding operations with identity operators, and additionally feed zeros across the skip connections. This is because the function of the skip connections is superfluous due to them not being able to feed a compressed representation of the input. We note that the performance of the ResNet can potentially be improved by allocating the number of channels different to the increasing and decreasing choice in a U-Net, but we have not explored this. We further train on `Shallow water` for 10 epochs evaluating on the test set, and on `Navier-Stokes` for approximately 400 K iterations evaluating on the validation set. This is due to significantly longer running times of the ResNets constructed as outlined above. We note that performance decreases very slowly after this point, and the trend of this experiment is unambiguously clear, not requiring extra training time.

The results in Table 8 show that preconditioning via the U-Net's self-similarity structure is a key reason for their empirical success. The Residual U-Net on multiple subspaces outperforms the ResNet

on a single subspace by a large margin. This also justifies the focus of studying preconditioning in this work.

Table 8: On the effect of subspace preconditioning vs. a plain ResNet. Quantitative performance of the (Haar Wavelet) Residual U-Net over multiple input and output subspaces compared to a ResNet on a single subspace, both trained on `Navier-Stokes` and `Shallow water`.

| Dataset | Neural architecture | # Params. | r-MSE ↓ |
|---------|--------------------|-----------|----------|
| **Navier-stokes** $128 \times 128$ | Residual U-Net (multiple subspaces) | 34.5 M | $\mathbf{0.0086 \pm 9.7 \cdot 10^{-5}}$ |
| | ResNet (one subspace) | 34.5 M | $0.3192 \pm 0.0731$ |
| **Shallow water** $96 \times 192$ | Residual U-Net (multiple subspaces) | 34.5 M | $\mathbf{0.3454 \pm 0.02402}$ |
| | ResNet (one subspace) | 34.5 M | $2.9443 \pm 0.2613$ |

### B.4.3 Ablation 3: On the importance of the wavelet transform in Multi-ResNets

In Table 9 we analyse the effect of different wavelets for the DWT we compute in $P_i$. We selected examples from different wavelet families, all of them being orthogonal. A good overview of different wavelets which can be straight-forwardly used to compute DWT can be found on this URL: `https://wavelets.pybytes.com/`. On `Navier-Stokes` we analyse the effect of choosing Daubechies 2 and 10 wavelets systematically over different random seeds. On `Shallow water` we explore many different Wavelet families generally without computing random seeds. We find that the Wavelets we choose for $P_i$ have some, but rather little impact on model performance. We lastly note that the downsampling operation has also been studied in [60] in the context of score-based diffusion models.

Table 9: On the importance of choosing different wavelets for the Discrete Wavelet Transforms (DWT) in Multi-ResNets. We report quantitative performance on the test set of `Navier-Stokes` and `Shallow water`.

| Dataset | Neural architecture | # Params. | r-MSE ↓ |
|---------|--------------------|-----------|----------|
| **Navier-stokes** $128 \times 128$ | Multi-ResNet w/ Haar wavelet DWT | 34.5 M | $\mathbf{0.0040 \pm 2 \cdot 10^{-5}}$ |
| | Multi-ResNet w/ Daubechies 2 DWT | 34.5 M | $0.0041 \pm 3 \cdot 10^{-5}$ |
| | Multi-ResNet w/ Daubechies 10 DWT | 34.5 M | $0.0068 \pm 1.1 \cdot 10^{-4}$ |
| **Shallow water** $96 \times 192$ | Multi-ResNet w/ Haar wavelet DWT | 34.5 M | $0.1493 \pm 0.0070$ |
| | Multi-ResNet w/ Daubechies 2 DWT | 34.5 M | $\mathbf{0.1081}$ |
| | Multi-ResNet w/ Daubechies 10 DWT | 34.5 M | $0.1472$ |
| | Multi-ResNet w/ Coiflets 1 DWT | 34.5 M | $0.1305$ |
| | Multi-ResNet w/ Discrete Meyer DWT | 34.5 M | $0.1237$ |

## B.5 On the importance of preconditioning in residual learning: Synthetic experiment

We begin by providing a second example with $w(v) = v^3$ in Figure 20, corresponding to Figure 2 in §1. Note that in comparison to the problem $w(v) = v^2$ in Figure 2, the effect of the two preconditioners swaps in Figure 20: Using $R^{\mathrm{pre}}(v) = v$ as the preconditioner produces an almost perfect approximation of the ground-truth function, while choosing $R^{\mathrm{pre}}(v) = |v|$ results in deteriorate performance. This is because $R^{\mathrm{pre}}(v) = v$ is a 'good' initial guess for $w(v) = v^3$, but a very 'poor' guess for $w(x) = v^2$, and vice versa for $R^{\mathrm{pre}}(v) = |v|$. This illustrates that the choice of using a good preconditioner is *problem-dependent*, and can be crucial to solve an optimisation problem at hand.

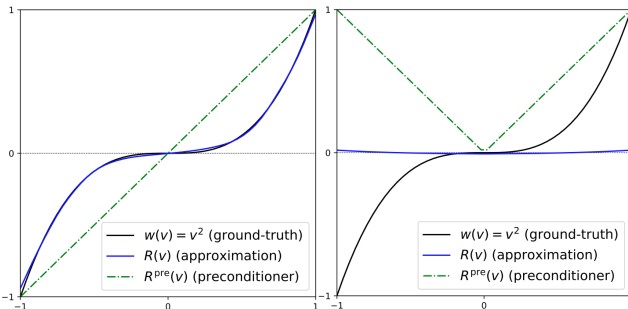

Figure 20: The importance of *preconditioning*. We learn the ground-truth functions $w(v) = \{v^3\}$ using a ResNet $R^{\mathrm{res}}(v) = R^{\mathrm{pre}}_\ell(v) + R(v)$ with preconditioners [Left] $R^{\mathrm{pre}}_1(v) = v$ and [Right] $R^{\mathrm{pre}}_2(v) = |v|$.

Having discussed the results of our synthetic experiment (see Figs. 2 and 20) which demonstrates the importance of preconditioning, we here provide further details on the experimental setting. We train a ResNet $R^{\mathrm{res}}(v) = R^{\mathrm{pre}}_\ell(v) + R(v)$ over a grid of values $\{(v_i, w_i)\}_{i=1}^N$ where $v_i \in [-1, 1]$, $w_i = w^{(k)}(v_i)$, $N = 50$, and $w^{(1)}(v) = v^2, w^{(2)}(v) = v^3, R^{\mathrm{pre}}_\ell(v) = v, R^{\mathrm{pre}}_\ell(v) = |v|$ depending on the experiment scenario indicated by the superscripts $(k, \ell)$, respectively. We do not worry about generalisation in this example and use the same train and test distribution. This functional relationship is easy to learn by any neural network $R$ of sufficient size, which would overshadow the effect of preconditioning. To construct a ResNet which is 'just expressive enough' to learn $w^{(k)}$, we constrain the expressivity of $R$ in two ways: First, we choose $R$ to be a small fully-connected network with 3 linear layers (input, hidden, output) with $[1, 20, 1]$ neurons, respectively, interleaved with the ReLU activation function. Second, inspired by invertible ResNets [61], we normalize the weights $\theta_j$ of $R$ such that $\|\theta_j\| < 1$ for all $j$ where $\|\cdot\|$ is the Frobenius norm, and repeatedly apply $R$ with $D = 100$ times via weight-sharing. In practice, $\|\theta_j\|$ will be close to 1, and as $\|\theta_j\|$ intuitively measures the 'step size' of a neural network, we constrain the neural network to '100 steps of unit length'. As an alternative, one could limit the maximum number of training steps, which would indirectly constrain the model's expressiveness. We note that our reported results are robust across a variety of hyperparameter combinations leading to qualitatively the same conclusions.

# C Background

## C.1 Related work

**U-Nets.** The U-Net [1] is a go-to, state-of-the-art architecture across various research domains and tasks. Among the most influential U-Net papers are U-Net++ [3] and Attention U-Net [4] for image segmentation, 3D U-Net [62] for volumetric segmentation, U-Nets in diffusion models, starting with the first high-resolution demonstration in DDPM [8], the nnU-Net [63, 64], a U-Net which automatises parts of its design, a probabilistic U-Net [65], conditional U-Nets [66], U-Nets for cell analysis detection and counting [67], and U-Nets for road extraction [68]. A large number of adaptations and variants of the U-Net exist. We refer to [69] for an overview of such variants in the context of image segmentation for which the U-Net was first invented. In particular, many U-Net papers use a ResNet architecture. [70] find that a key improvement for the seminal U-Net [1] is to use residual blocks instead of feed-forward convolutional layers. Beyond their use on data over a squared domain, there exist custom implementations on U-Nets for other data types, for instance on the sphere [17], on graphs [18, 71] or on more general, differentiable 3D geometries [72]. However, we note that a framework which unifies their designs and details the components for designing U-Nets on complicated data types and geometries is lacking. This paper provides such a framework.

Our work directly builds on and is motivated by the paper by Falck & Williams [20] which first connected U-Nets and multi-resolution analysis [73]. This paper showed the link between U-Nets and wavelet approximation spaces, specifically the conjugacy of Haar wavelet projection and average pooling in the context of U-Nets, which our work crucially relies on. The design of U-Nets and their connection to wavelets has also been studied in [25, 26]. Falck & Williams further analysed the regularisation properties of the U-Net bottleneck under specific assumptions restricted to the analysis of auto-encoders without skip connections, and argued by recursion what additional information is carried across each skip without this being rigorously defined. Our work in this paper is however augmenting their work in various ways. We list five notable extensions: First, we provide a unified definition of U-Nets which goes beyond and encompasses the definition in [20], and highlights the key importance of preconditioning in U-Nets as well as their self-similarity structure. Importantly, this definition is not limited to orthogonal wavelet spaces as choices for $\mathcal{V}$ and $\mathcal{W}$, hence enabling the use of U-Nets for a much broader set of domains and tasks. Second, based on this definition, our framework enables the design of novel architectures given a more thorough understanding of the various components in a U-Net. This is demonstrated on the elliptic PDE problem and data over a triangular domain. Third, we analyse the usefulness of the inductive bias of U-Nets in diffusion models, a novel contribution. Fourth, our experiments with Multi-ResNets and multi-resolution training and sampling, as well as on triangular data are novel. Fifth, [20] focusses particularly on the application of U-Nets in hierarchical VAEs, which this work is not interested in in particular. In summary, while [20] provided crucial components and ideas in U-Nets, our work is focused on a framework for designing and analysing general U-Nets, which enables a wide set of applications across various domains.

**Miscellaneous.** We further briefly discuss various unrelated works which use similar concepts as those in this paper. Preconditioning, initialising a problem with prior information so to facilitate solving it, is a widely used concept in optimisation across various domains [74, 75, 76, 77, 13, 14]. In particular, we refer to its use in multi-level preconditioning and Garlekin methods [78, 79, 80]. Preconditioning is also used in the context of score-based diffusion models [81]. Most notably, preconditioning is a key motivation in the seminal paper on ResNets [15]. While preconditioning is a loosely defined term, we use it in the context of this literature and its usage there.

The concept of 'dimensionality reduction' is widely popular in diffusion models beyond its use in U-Nets. For instance, Stable Diffusion and Simple Diffusion both perform a diffusion in a lower-dimensional latent space and experience performance gains [11, 60]. Stable Diffusion in particular found that their model learns a "low-dimensional latent space in which high-frequency, imperceptible details are abstracted away" [11]. It is this intuition that the U-Net formalises. Simple Diffusion also features a multi-scale loss resembling the staged training in Algorithm 1. Another paper worth pointing out learns score-based diffusion models over wavelet coefficients, demonstrating that wavelets and their analysis can be highly useful in diffusion models [42].

## C.2 Hilbert spaces

A Hilbert space is a vector space $W$ endowed with an inner-product $\langle \cdot, \cdot \rangle$ that also induces a complete norm $\|\cdot\|$ via $\|w\|^2 = \langle w, w \rangle$. Due to the inner-product, we have a notion of two vectors $w_1, w_2 \in W$ being orthogonal if $\langle w_1, w_2 \rangle = 0$.

In Section 3 we have paid special attention to two specific Hilbert spaces: $L^2([0,1])$ and $\mathcal{H}_0^1([0,1])$.

1. the space $L^2([0,1])$ has as elements square-integrable functions and has an inner-product given by

$$\langle f, g \rangle_{L^2} = \int_0^1 f(x)g(x)dx; \text{ and,} \tag{34}$$

2. the space $\mathcal{H}_0^1([0,1])$ has as elements once weakly differentiable functions which vanish at both zero and one, with inner-product given by

$$\langle f, g \rangle_{\mathcal{H}_0^1} = \int_0^1 f'(x)g'(x)dx, \tag{35}$$

where $f', g' \in L^2([0,1])$ are the weak derivatives of $f$ and $g$ respectively.

If we have a sequence $\{\phi_k\}_{k=0}^\infty$ of elements of $W$ which are pairwise orthogonal and span $W$, then we call this an orthogonal basis for $W$. Examples of orthogonal bases for both $L^2([0,1])$ and $\mathcal{H}_0^1([0,1])$ are given in Section C.3.

## C.3 Introduction to Wavelets

Wavelets are refinable basis functions for $L^2([0,1])$ which obey a self-similarity property in their construction. There is a 'mother' and 'father' wavelet $\phi$ and $\psi$, of which the children wavelets are derivative of the mother wavelet $\phi$. For instance, in the case of Haar wavelets with domain $[0,1]$ we have that $\psi(x) = 1$ and the mother and children wavelets given by

$$\phi_{k,j}(x) = \phi(2^k x + j/2^k), \qquad \phi(x) = 1_{[0,1/2)}(x) - 1_{[1/2,1]}(x).$$

Under the $L^2$-inner product, the family $\{\phi_{k,j}\}_{j=0,k=1}^{2^j,\infty}$ forms an orthogonal basis of $L^2([0,1])$. Further, if we define the functions

$$\tilde{\phi}_{k,j}(x) = \int_0^x \phi_{k,j}(y)dy, \tag{36}$$

then each of these functions is in $\mathcal{H}_0^1([0,1])$, and is further orthogonal with respect to the $\mathcal{H}_0^1$-inner-product, bootstrapped from the orthogonality of the Haar wavelets.

The discrete encoding of the Haar basis can be given by the kronecker product

$$H_i = \begin{pmatrix} 1 & 1 \\ 1 & -1 \end{pmatrix} \otimes H_{i-1}, \qquad H_0 = \begin{pmatrix} 1 & 1 \\ 1 & -1 \end{pmatrix}. \tag{37}$$

To move between the pixel basis and the Haar basis natural to average pooling, we use the mapping defined on resolution $i$ by $T_i : V_i \mapsto V_i$ through

$$T_i(v_i) = \Lambda_i H_i v_i, \tag{38}$$

where $H_i$ is the Haar matrix above and $\Lambda_i$ is a diagonal scaling matrix with entries

$$(\Lambda_i)_{k,k} = 2^{-i+j-1}, \quad \text{if } k \in \{2^{j-1}, \dots, 2^j\}. \tag{39}$$

To get this, we simply identify how to represent a piecewise constant function from its pixel by pixel function values to be the coefficients of the Haar basis functions. For example, in one dimension for an image with four pixels we can describe the function as the weighted sum of the four basis functions [Right], that take values $\pm 1$ or zero. The initial father wavelet is set to be the average of the four pixel values, and the coefficients of the mother and children wavelets are chosen to be the local deviance's of the averages from these function values, as seen in Figure 21.

For further details on wavelets, we refer to textbooks on this topic [73, 29].

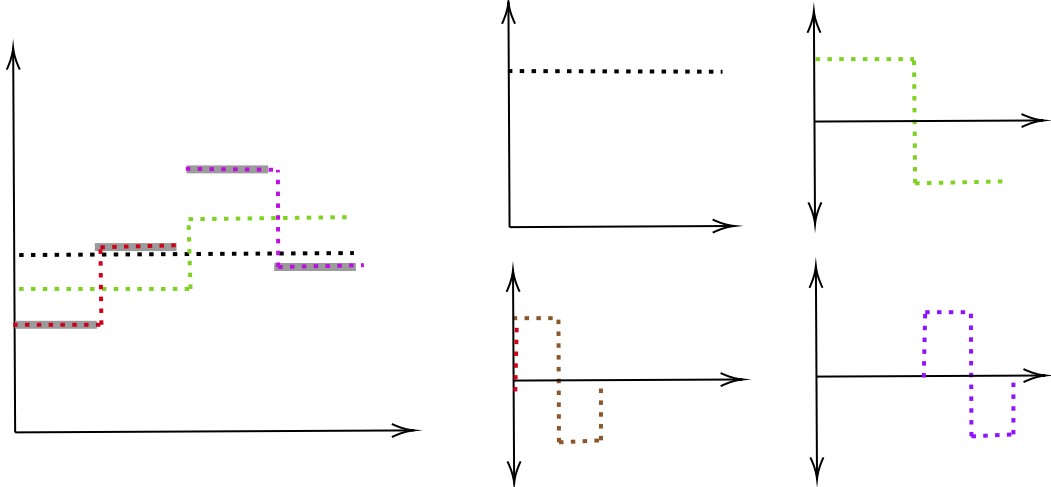

Figure 21: Modelling a 1D image with four pixel values as the weighted sum of Haar wavelet frequencies. The coefficients are such that the local averages of pixel values give the Haar wavelet at a lower frequency, hence average pooling is conjugate to basis truncation here.

## C.4   Images are functions

An image, here a gray-scale image with squared support, can be viewed as the graph of a function over the unit square $[0,1]^2$ [20]. We visualise this idea in 22, referring to [20, Section 2] and [19] for a more detailed introduction. Many other signals can likewise be viewed as It is hence natural to construct our U-Net framework on function spaces, and have $E_i$ and $D_i$ be operators on functions.

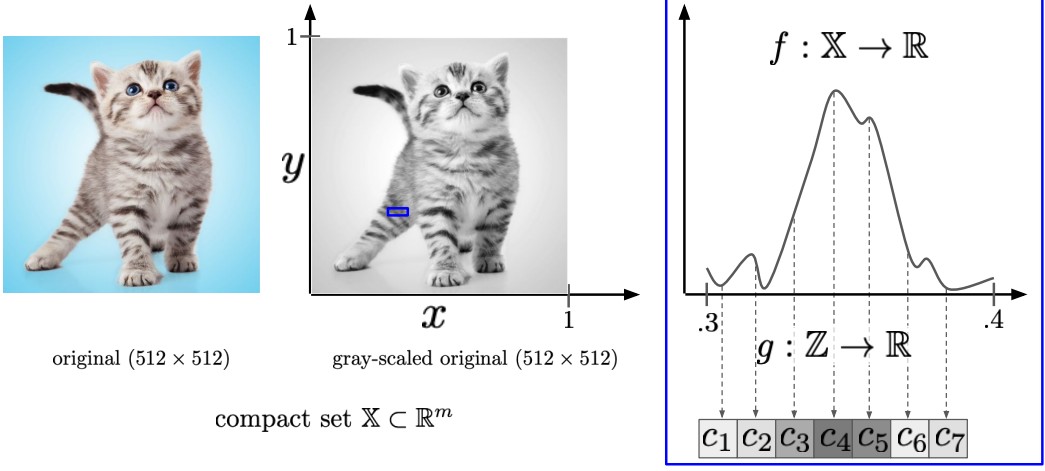

Figure 22: Images modelled as functions.

## D   Code, computational resources, datasets, existing assets used

**Code.**   We provide our code base as well as instructions to reproduce the key results in this paper under MIT License at https://github.com/FabianFalck/unet-design. Our code base uses code from four Github repositories which are subfolders. For our diffusion experiments on MNIST and MNIST-Triangular, we directly build on top of the repository https://github.com/JTT94/torch_ddpm. For our diffusion experiments on CIFAR10, we directly build on top of the repository https://github.com/w86763777/pytorch-ddpm. For our PDE experiments on Navier-Stokes and Shallow water, we use the repository https://github.com/microsoft/pdearena [6]. For our image semgentation experiments on WMH, we are inspired by the repository

https://github.com/hongweilibran/wmh_ibbmTum [51], but write the majority of code ourselves. We note that the MIT License only pertains to the original code in our repository, and we refer to these four repositories for their respective licenses.

We extended each of these repositories in various ways. We list key contributions below:

- We implemented several Residual U-Net architectures in the different repositories.
- We implemented Multi-ResNets in each repository.
- We implemented the staged training Algorithm (see Algorithm 1), as well as its strict version which freezes parameters.
- We implemented logging with weights & biases, as well miscellaneous adjustments for conveniently running code, for instance with different hyperparameters from the command line.

**Datasets.** In our experiments, we use the following five datasets: MNIST [47], MNIST-Triangular, a custom version of MNIST where data is supported on a triangle rather than a square, CIFAR10 [48], Navier-stokes, Shallow water [49], and the MICCAI 2017 White Matter Hyperintensity (WMH) segmentation challenge dataset [50, 51]. These five datasets—with the exception of MNIST-Triangular—have in common that data is presented over a square or rectangular domain, possibly with several channels, and of varying resolutions. We refer to the respective repositories above where these datasets have already been implemented. For MNIST-Triangular, we provide our custom implementation and dataset class as part of our code base, referring to Appendix B.3 on how it is constructed. It is also worth noting that Navier-Stokes and Shallow Water require considerable storage. On our hardware, the two datasets take up approximately 120 GB and 150 GB unzipped, respectively.

**Computational resources.** This work made use of two computational resources. First, we used two local machines with latest CPU hardware and one with an onboard GPU for development and debugging purposes. Second, we had access to a large compute cluster with A100 GPU nodes and appropriate CPU and RAM hardware. This cluster was shared with a large number of other users.

To reproduce a single run (without error bars) in any of the experiments, we provide approximate run times for each dataset using the GPU resources: On MNIST and MNIST-Triangular, a single run takes about 30 mins. On CIFAR10, a single run takes several days. On Navier-Stokes and Shallow water, a single run takes approximately 1.5 days and 1 day, respectively.

**Existing assets used.** Our code base uses the following main existing assets: Weights&Biases [82] (MIT License), PyTorch [83] (custom license), in particular the torchvision package, pytorch_wavelets [84] (MIT License), PyWavelets [85] (MIT License), pytorch_lightning [86] (Apache License 2.0), matplotlib [87] (TODO), numpy [88] (BSD 3-Clause License), tensorboard [89] (Apache License 2.0), PyYaml [90] (MIT License), tqdm [91] (MPLv2.0 MIT License), scikit-learn and sklearn [92] (BSD 3-Clause License), and pickle [93] (License N/A). We further use Github Copilot for the development of code, and in few cases use ChatGPT [94] as a writing aid.

