# OpenReview forum: "A Unified Framework for U-Net Design and Analysis"
_NeurIPS.cc/2023/Conference — NeurIPS 2023 poster_

### Official Review · Reviewer_go4z · 2023-06-19

**Soundness:** 3 good
**Presentation:** 2 fair
**Contribution:** 3 good
**Rating:** 7
**Confidence:** 3

**Summary:**

This paper proposes a formal definition of U-nets, a crucial building block of modern deep learning pipelines such as diffusion models.
Thanks to this definition it is possible to both get theoretical results explaining some of the u-nets behaviours, and generalize u-nets to settings more exotic than the 2D images.
Experiments are then conducted on various modalities to validate the theory.

**Strengths:**

- I agree with the main selling point of the paper : U-Nets have been under studied, especially in the context of diffusion models, and it is more than necessary to put an end to that.
- I think the formalism effort presented in this work is great, and it was indeed, to the best of my knowledge, lacking.
- the diversity of the modalities studied showcases the potential impact of better understanding U-Nets

**Weaknesses:**

- **inductive bias**: theorem 2 is the cornerstone of the theoretical explanation of this work as to why U-Nets are effective in diffusion. Yet I think that it has 2 problems. First, the result is about the Haar decomposition of the diffusion process, and therefore it applies to all models that have a multiscale structure with average pooling, not specifically U-Nets. Second, and in my view much more problematic, diffusion pipelines often use an "epsilon-type" prediction framework, in which the role of the U-Net is not to predict the denoised image (or signal in general), but rather the noise.
There are multiple examples of this that can be found either in the examples of the diffusers library of HuggingFace or in the seminal papers of Ho and Song:
    - in the unconditional generation example: https://github.com/huggingface/diffusers/blob/main/examples/unconditional_image_generation/train_unconditional.py#L576, when prediction type is "epsilon", the loss is computed as the MSE between the output of the model and the noise, and one can see that this is the default mode (L229), not changed in the way the script is called (https://github.com/huggingface/diffusers/blob/main/examples/unconditional_image_generation/README.md)
    - in the text-to-image generation example, the prediction type is taken by default from the noise scheduler which has it by default has "epsilon" (https://github.com/huggingface/diffusers/blob/716286f19ddd9eb417113e064b538706884c8e73/src/diffusers/schedulers/scheduling_ddpm.py#L121), not changed in the way the script is called (https://github.com/huggingface/diffusers/blob/main/examples/text_to_image/README.md)
    - In [7], algorithm 1 clearly shows that it's the noise that is learned
    - In the implementation of [34], the noise `z` is used used in the MSE with the model output (https://github.com/yang-song/score_sde/blob/main/losses.py#L115)
As a consequence I think the theoretical result does not cover the practice. Indeed, it is said "This inductive bias enables the encoder and decoder networks to focus on the signal on a low enough frequency which is not dominated by noise." while the network's role is precisely to focus on subspaces which feature the noise in order to return it. While I understand that some pipelines still use a "sample" type prediction, I think that the understanding of the U-Net inductive bias is not understood if it doesn't take into account both types of predictions, especially if the explanation of one type is contrarian to the other type.
- **clarity and space use**: I think that some sections of the paper are very unclear, in part because some essential parts were left out in the appendix. For example, Algorithm 1 for staged training should be in the core paper. Similarly the practical construction of a U-Net that guarantees boundary conditions should be in the core paper. For now, we just get an idea of what the decoder subspaces should look like, but not how the decoder should be designed in order to achieve these subspaces. Right now it is not clear to me how these decoders are designed. A last example (although there are some more) is that the formal definition of the U-Net is hard to grasp at first read. There should be a correspondence between the elements introduced and the typical 2D image segmentation U-Net architecture.
- **prior work comparison/mention**: I noticed two parts that needed more references to prior work/comparison to it. First, the connection between U-Nets and wavelets has been a topic of many works among which [A, B]. Second, diffusion on complicated geometries including manifolds has been recently covered by works such as [C-F]. For both of these, mentioning as well as highlighting the differences would be super important.

*Minor*:
- To me preconditioning means finding a matrix close to the Hessian inverse in an optimization procedure in order to better take into account the geometry of the problem when optimizing, not finding a good initialization. Here it seems that preconditioning is more used to describe a good initialization rather than taking into account geometric information.
- "It also explains why in practice, encoders are often chosen significantly smaller than the decoder [26]". I am not sure about this fact, so I think it needs much more evidence (see for example in the diffusers library https://github.com/huggingface/diffusers/blob/main/src/diffusers/models/unet_2d.py#L198
the decoder only has one more layer than the encoder). Also [26] is about VAEs, not U-Nets.



[A]: Ramzi, Z., Michalewicz, K., Starck, JL. et al. Wavelets in the Deep Learning Era. J Math Imaging Vis 65, 240–251 (2023).

[B]: Liu, P., Zhang, H., Zhang, K., Lin, L., & Zuo, W. (2018). Multi-level wavelet-CNN for image restoration. In Proceedings of the IEEE conference on computer vision and pattern recognition workshops (pp. 773-782).

[C]: De Bortoli, V., Mathieu, E., Hutchinson, M., Thornton, J., Teh, Y. W., & Doucet, A. (2022). Riemannian score-based generative modeling. arXiv preprint arXiv:2202.02763.

[D]: Fishman, N., Klarner, L., De Bortoli, V., Mathieu, E., & Hutchinson, M. (2023). Diffusion Models for Constrained Domains. arXiv preprint arXiv:2304.05364.

[E]: Huang, C. W., Aghajohari, M., Bose, J., Panangaden, P., & Courville, A. C. (2022). Riemannian diffusion models. Advances in Neural Information Processing Systems, 35, 2750-2761.

[F]: Liu, L., Ren, Y., Lin, Z., & Zhao, Z. (2022). Pseudo numerical methods for diffusion models on manifolds. arXiv preprint arXiv:2202.09778.


**Questions:**

- What is meant by self-similarity?
- In the code, for MNIST experiment, a network called GNet is used rather than a U-Net : what is it and does it fit with the experiments reported in the paper?
- When deriving Theorem 2, it seems that each subspace W_i is an image subspace. However, in practice the U-Net acts on feature spaces, i.e. spaces with much more feature channels than the image itself, how would you take that into account?

---

> ### Author Rebuttal · Authors · 2023-08-09
>
> We thank the reviewer for their thoughtful and detailed review. We are thankful for the appreciation of the formalism of our theoretical framework which contributes to an understudied area, among other strengths. We respectfully would like to point out that we disagree with the concerns on Theorem 2 which seem to be the reviewer’s main reason for discontent.
>
> > “First, the result is about the Haar decomposition of the diffusion process, and therefore it applies to all models that have a multiscale structure with average pooling, not specifically U-Nets.”
>
> Theorem 2 analyses the effect of average pooling (wavelet projection [4]) on data under a forward diffusion process. It is worth noting that Theorem 2 does not require or mention the concept of a U-Net per se, but is consequential to the model design of diffusion models. We appreciate your insight, which highlights the theorem's broader applicability to various architectures leveraging a multiscale structure with average pooling. We don’t think this is a “problem” of Theorem 2, it rather shows its strength and generality beyond U-Nets.
>
> The reviewer’s description of "multiscale structure with average pooling" captures what many might view as a U-Net's essence. This is just a curious example that the lack of a formal definition in the community necessitates our introduction of Definition 1. This definition serves not just as a concise representation, but also facilitates the deep dive into U-Nets' characteristics, from their recursive patterns (2.1) and scaling laws (2.2) to their relationship with ResNets (2.3).
>
> > “Second, [...] diffusion pipelines often use an "epsilon-type" prediction framework, in which the role of the U-Net is not to predict the denoised image (or signal in general), but rather the noise. [...] As a consequence I think the theoretical result does not cover the practice.”
>
> We respectfully disagree with this main concern on Theorem 2, and would like to find an agreement with the reviewer on this point.
>
> First, we would like to note that both the “epsilon-type” (noise as output) and the “sample-type” (image as output) solve *the same task*, namely to separate signal from noise. While in both frameworks the network receives image and noise $Im + \epsilon$ as input, the “epsilon-type” framework achieves this task by subtracting the signal $Im$, while the “sample-type” framework subtracts the noise $\epsilon$.
>
> The fundamental quantity of interest which governs how challenging it is to solve this task of separating signal and noise (in both formulations) is the signal-to-noise ratio between $Im$ and $\epsilon$. In Theorem 2, we demonstrate that the high-frequency components of an image have their signal-to-noise ratio dominated by noise exponentially faster than the low-frequency components. Importantly, a U-Net architecture with average pooling is passing as input to the decoder the image plus noise at the highest resolution, and the low frequency details that are dominated by signal on lower resolutions. This means that apriori to training the network, we have fed as input the image plus noise, and a first approximation of the image without noise. As the network learns to separate noise from signal, these two inputs yield a useful inductive bias for the signal and noise separation task in light of Theorem 2.
>
> While not entirely certain, we believe the confusion might arise from the following argument. During average pooling, the high-resolution Haar wavelet spaces, which are dominated by noise (Theorem 2), get discarded. However, in the  "epsilon-type" formulation, it may be unclear how the U-Net can now predict the noise, if ‘we just removed the noise’ via average pooling (more precisely, those subspaces dominated by noise). Should this indeed be the source of confusion, we can hopefully resolve this easily. The responsibility for ‘predicting the noise’ does not solely lie on the encoder. Instead, it's the decoder that uses the encoder's signal through skip connections. The encoder primarily functions as an ‘information compressor’ for the input data. The decoder utilises this condensed input to generate an output, which could be either the 'sample' or the 'epsilon' (noise). In either case, a lower-frequency, low-noise signal serves as valuable compressed information that aids the decoder in generating its output.
>
>  > ”I noticed two parts that needed more references to prior work/comparison to it [...] mentioning as well as highlighting the differences would be super important.”
>
> U-Nets and Wavelets: We appreciate the reviewer's references, especially [A], and will include [A,B] in our discussion of Haar subspace U-Nets. While these works highlight wavelet-U-Net connections, our Definition 1 unifies diverse wavelet bases and generalises to other bases. Our empirical study of Multi-ResNets, replacing the U-Net's encoder with wavelet projections, adds novel insight. This demonstrates the broader connection between U-Nets and wavelets [5].
>
> Diffusions over Complex Geometries: U-Nets achieve state-of-the-art results in learning diffusion models for image-based tasks [9]. Yet, their performance can waver when modelling functions on non-Euclidean spaces. Notably, recent research into diffusion models on Riemannian manifolds [C-F] has favoured simple, fully-connected networks over U-Nets or modifications thereof.
>
> Our unified framework bridges this gap by unveiling previously overlooked potential in neural architecture. Our work is hence complementary to [C-F], focussing on the neural architecture. It paves the way for U-Net adaptations tailored to complex geometries, including CW-complexes and manifolds by integrating data topology into the U-Net itself. This lays the foundation for neural architectures that suit diffusion models on manifolds.
>
> - - -
>
> Lastly, we refer to our response to all authors for a discussion of other points in this review.

---

> > ### Comment · Reviewer_go4z · 2023-08-11
> > **Thanks for your answers**
> >
> > Before answering, I would like to thank the authors for engaging in the review process in what I view as a very positive and intellectually honest manner.
> > I think that aside from the 2nd point about theorem 2, I don’t have major blockers left, granted the improvements mentioned in the rebuttal are implemented (which I don’t think is a big issue).
> >
> > *1st point on Th. 2., about specificity*: But then how does this result relate to e.g. the one by Donoho in “Denoising by soft thresholding” (1995)? He also looked at the variance of the noise in wavelet coefficients.
> >
> > *2nd point on Th. 2., about epsilon prediction*: I think the clarification that the decoder can focus on the high frequency is important, and to me it sounds contradicting with what was written in the paper “This inductive bias enables the encoder and decoder networks to focus on the signal on a low enough frequency which is not dominated by noise.”, but maybe it’s because I am understanding the word “focus” in this context in the wrong way. Still I think this demands clarification.
> >
> > Further, my main take-away from this response is that the inductive bias of the U-net basically (and maybe I am simplifying too much here) is that it has the ability to provide “a lower-frequency, low-noise [version of the] signal”. I don’t understand then how we can understand that this provides a good inductive bias, i.e. is “valuable compressed information that aids the decoder in generating its output”. To me this is the crux of the matter, rather than showing that the U-net’s encoder can extract this information.
> >
> > I realize that my problem is not necessarily on Theorem 2., which I see as valid (not having checked the proof thoroughly but it matches what I have seen in the past in wavelet works), but rather on the interpretation made thereafter.
> >
> > In the abstract it says "In diffusion models, our framework enables us to identify that high- frequency information is dominated by noise exponentially faster, and show how U-Nets with average pooling exploit this.". While I agree that the first part of this sentence is backed up in the text, to me the second part is not as I explain above.

---

> > > ### Author Response · Authors · 2023-08-13
> > > **Reply to: 'Thanks for your answers' (1/3)**
> > >
> > > We very much share the positive sentiment about this useful and enlightening discussion with the reviewer. We are thankful for the questions the reviewer is raising which allow us to improve and produce a clearer, more accurate explanation of our results. We are glad we could resolve the reviewer’s concerns in most points through the explanations and improvements provided, and would like to discuss the remaining questions below. We would be thankful if you could consider the outcome of this discussion towards the final recommendation and score of your review.
> > >
> > >
> > > > “1st point on Th. 2., about specificity: But then how does this result relate to e.g. the one by Donoho in “Denoising by soft thresholding” (1995)? He also looked at the variance of the noise in wavelet coefficients.”
> > >
> > > We appreciate the reviewer for highlighting this relevant article. We recognise a notable similarity between the message of our Theorem 2 and the analysis of coefficient decay in the pertinent Besov space in [Donoho1]. Under the forward diffusion process analysed in Theorem 2, the signal of our data predominantly concentrates on the low-frequency wavelet coefficients. In light of [Donoho1], this observation can also be articulated as: when examining the appropriate sequence of wavelet coefficients in a sequence space, the high-frequency coefficients, in expectation, tend to zero considerably faster than their low-frequency counterparts. Therefore, restating Theorem 2 in the context of the Besov space would indeed enhance clarity, especially in elucidating how this property can be generalised to other geometries, given that all analysis is anchored in the Besov space. We propose to incorporate this analysis into the appendix of our manuscript, with reference to the given article, complemented by a more generalised statement of Theorem 2 for users working with U-Nets in complicated geometries.
> > >
> > > Another intriguing idea would be to utilise this sequence space information from a specific dataset to define the noise schedule of a diffusion process and to determine the size of each decoder block based on these coefficients. While delving deeper into this would require additional research, we are confident that presenting this information in a straightforward manner could be an interesting extension towards better U-Net designs *tailored to individual datasets*. We believe that presenting Theorem 2 in this expanded manner, coupled with our general U-Net framework, marks an initial step towards such research. The crux here lies in understanding how this sequence space decays, because as demonstrated in [Donoho1], this can provide insights into our model's design.
> > >
> > > The reviewer may also be referring to the wavelet shrinkage procedure on page 3 first proposed in (Donoho, 1992). In step (2), wavelet coefficients are translated and then thresholded (the operator $(\cdot)_+$ is in our understanding the ReLU function) which effectively results in a ‘continuous shrinkage to zero’ of wavelet coefficients, where some coefficients are indeed set to zero. This is in contrast to the ‘hard projection’ in average pooling, where coefficients are either their identity or set to zero. It is also worth noting that this procedure makes the assumption that the reconstruction is at least as smooth as the ground-truth function that shall be approximated (see (1.3)), an assumption which Theorem 2 does not make.
> > >
> > > On a final note: [Donoho1] is a long paper with many results and we admit that it is slightly challenging at present. If we should take into consideration another specific result from this paper that we have not discussed above, we would appreciate it if the reviewer could point us to it. Lastly, we would like to point out that there are two versions of this paper. We carefully read the version cited below [Donoho1].
> > >
> > > We thank the reviewer for this very interesting reference to relevant literature, and the useful discussion on it.
> > >
> > > [Donoho1] https://web.stanford.edu/dept/statistics/cgi-bin/donoho/wp-content/uploads/2018/08/denoiserelease3.pdf

---

> > > > ### Author Response · Authors · 2023-08-13
> > > > **Reply to: 'Thanks for your answers' (2/3)**
> > > >
> > > > Below, we will respond to several comments coherently.
> > > >
> > > >
> > > > > “2nd point on Th. 2., about epsilon prediction: I think the clarification that the decoder can focus on the high frequency is important, and to me it sounds contradicting with what was written in the paper ‘This inductive bias enables the encoder and decoder networks to focus on the signal on a low enough frequency which is not dominated by noise.’, but maybe it’s because I am understanding the word ‘focus’ in this context in the wrong way. Still I think this demands clarification.
> > > >
> > > > To resolve this confusion, we would like to cite and explain the two sentences following the cited sentence above: “[...] As the subspaces are coupled via preconditioning, the U-Net can learn the signal which is no longer dominated by noise, added on each new subspace. This renders U-Nets a computationally efficient choice in diffusion models [...].” Please allow us to explain this with reference to Sec 2.
> > > >
> > > > Say a U-Net $U_i$ is making an approximation on resolution $i$. Then, the decoder $D_i$ on resolution $i$ and output space $W_i$ is making an approximation *preconditioned* (i.e. receiving as input) the output of the U-Net $U_{i-1}$ on resolution $i-1$, which is also the output of the decoder on resolution $i-1$ (see Eq. (1)). Hence, the decoder $D_i$ on the resolution we wish to approximate receives as input a ‘lower resolution approximation’ of the output it wishes to produce, and can in this sense focus on the added information (i.e. the high-frequency details) relative to what it is preconditioned on (i.e. the ‘lower-resolution approximation’ of $U_{i-1}$. The encoder $E_i$ facilitates this via its skip connection by providing a ‘lower-dimensional, low-noise input’, as the reviewer correctly stated, or more precisely, a representation in a changed basis thereof. This input is useful for the learning task of the decoder $D_i$ regardless of whether it is outputting noise (epsilon-type) or signal (sample-type) as we outlined in our rebuttal. We hope this provides an explanation of what we mean by the decoder being able to “focus on high frequency” details in the paper.
> > > >
> > > > What we intended to express with the sentence quoted by the reviewer is that the encoder and decoder networks can focus their ‘learning task’ on only the frequency information added on a certain resolution, in light of preconditioning. The decoder produces the actual approximation of the output on resolution $i$, preconditioned on the previous resolution’s approximation. Crucially, it also requires the encoder $E_i$’s output as input. If we choose $P_i$ as average pooling, as a first approximation of the encoder, we project to lower-resolution subspaces, because average pooling is conjugate to projection in a Haar wavelet basis (see Theorem 2 in [5] for a detailed explanation), which is also the basis of choice in Theorem 2. According to Theorem 2 in the submission, these subspaces correspond to ‘low noise, high signal’. The encoder $E_i$ can then learn a change of basis thereof and pass it to the decoder. In this sense, both the encoder and decoder can focus on a “low enough frequency which is not dominated by noise”, enabled via average pooling.
> > > >
> > > > Based on this discussion, we suggest to reformulate the quoted sentence to: ***‘This inductive bias enables the decoder to have access to signal-dominated, low-frequency information, which is not dominated by noise.’*** We thank the reviewer in particular for enabling this improvement.

---

> > > > > ### Author Response · Authors · 2023-08-13
> > > > > **Reply to: 'Thanks for your answers' (3/3)**
> > > > >
> > > > > > “Further, my main take-away [...] is that the inductive bias of the U-net [...] it has the ability to provide ‘a lower-frequency, low-noise [version of the] signal’. I don’t understand then how we can understand that this provides a good inductive bias, i.e. is ‘valuable compressed information that aids the decoder in generating its output’. [...]”
> > > > >
> > > > > > “In the abstract it says ‘In diffusion models, our framework enables us to identify that high- frequency information is dominated by noise exponentially faster, and show how U-Nets with average pooling exploit this.’. While I agree that the first part of this sentence is backed up in the text, to me the second part is not as I explain above.”
> > > > >
> > > > > We hope that given our response above, the usefulness of average pooling, it being a beneficial inductive bias has become more apparent. Another way of putting it is the following argument: our goal in diffusion models is to separate the signal from the noise, both in the epsilon-type and sample-type formulation. ***If we perform average pooling many times, we get lower-resolution coefficients where the signal-to-noise ratio is high, i.e. where it is easy to separate the signal from the noise. This is exactly what we mean by ‘beneficial inductive bias’: The network’s architecture facilitates solving the approximation task.*** We can easily and computationally efficiently make a good approximation on the lower-resolution, and one resolution after the other (via preconditioning) focus on higher resolutions where separating the signal from noise is significantly harder.
> > > > >
> > > > > In light of this discussion, we acknowledge that the interpretation of Theorem 2 could be clearer and extended. Towards a potential camera-ready version, we will use the above discussion to clarify Theorem 2, and very much thank the reviewer for pointing this need out to us.
> > > > >
> > > > >
> > > > > > “I realize that my problem is not necessarily on Theorem 2., which I see as valid (not having checked the proof thoroughly but it matches what I have seen in the past in wavelet works), but rather on the interpretation made thereafter.”
> > > > >
> > > > > We are glad that we seem to be able to resolve some of these questions on the applicability of Theorem 2 through this discussion, and agree that the remaining questions are mostly on the interpretation of Theorem 2 (which is crucially important!). While we tried very hard, explaining mathematical statements in words is always challenging and can only get epsilon-close to the truth. Hence we very much appreciate this discussion as these insights are far from straight-forward, and are happy to clarify any further questions, if required.

---

> > > > > > ### Comment · Reviewer_go4z · 2023-08-14
> > > > > >
> > > > > > I repeat again here that I am extremely happy with the way the conversation is happening, and I thank the authors ocne more for engaging in such a positive way.
> > > > > >
> > > > > > I think my comment on the first point has been well received.
> > > > > >
> > > > > > I am happy with how the theorem 2 is planned to be discussed from now on in the camera-ready version of the paper.
> > > > > >
> > > > > > For all of these reasons I am increasing my score to 7, Accept.
> > > > > > The reason I am not giving an even higher score is because I think that Theorem 2 shares too many similarities with the existing result by Donoho.
> > > > > >
> > > > > > I have 2 last points:
> > > > > > - I think it would be valuable to add in a comment (maybe in appendix as an extended discussion) on how max pooling can also extract low resolution information (can it?).
> > > > > > - Another interesting reference might be: “DEEP CONVOLUTIONAL FRAMELETS: A GENERAL DEEP LEARNING FRAMEWORK FOR INVERSE PROBLEMS“ by Ye et al. 2018.

---

> > > > > > > ### Author Response · Authors · 2023-08-15
> > > > > > > **Reply to: Official Comment by Reviewer go4z**
> > > > > > >
> > > > > > > We would like to thank the reviewer for the valuable and profound discussion. We truly appreciate the reviewer’s efforts and the questions raised, particularly on Theorem 2, which contribute to and allow us to provide an improved, clarified version of our submission in a potential camera-ready version.
> > > > > > >
> > > > > > > We thank the reviewer for bringing up the topic of max pooling and we will comment how this fits within our framework in an updated appendix as well as discuss the relevant paper suggested in the related work accordingly.
> > > > > > >
> > > > > > > Lastly, we are grateful to the reviewer for viewing to accept our paper on the basis of this discussion. We have found this interaction to be very beneficial to the improvement of our work and are thankful to the reviewer in this regard.

---

### Official Review · Reviewer_ndAy · 2023-06-25

**Soundness:** 2 fair
**Presentation:** 2 fair
**Contribution:** 2 fair
**Rating:** 5
**Confidence:** 3

**Summary:**

This paper proposes a framework for designing and analyzing general UNet architectures. Theoretical results are presented that can characterize the role of encoder and decoder in UNet and conjugacy to ResNets is pointed out via preconditioning. Furthermore, this paper proposes Multi-ResNets, UNets with a simplified, wavelet-based encoder without learnable parameters. In addition, this paper presents how to design novel UNet architectures which can encode function constraints, natural bases, or the geometry of the data. Experiments of Multi-ResNets on image segmentation, PDE surrogate modeling, and generative modeling with diffusion model are conducted and demonstrate competitive performance compared to classical UNet.

**Strengths:**

- Provide the mathematical definition of UNet, which enable identifying self-similarity structure, high-resolution scaling limits, and conjugacy to ResNets via preconditioning.
- Based on theoretical analysis, authors propose Multi-ResNets, a novel class of UNet with no learnable parameters in its encoder.
- Multi-ResNets achieves competitive or superior results compared to a classical U-Net in PDE modeling, image segmentation, and generative modeling with diffusion models.


**Weaknesses:**

- The presentation of the paper is not smooth and clear, which needs significant improvement. For example, the authors claim to propose a unified framework for UNet design, but I don’t see how this model can help researchers design specific UNet for different tasks. It’s more like the authors improve UNet on three chosen tasks and provide some theoretical analysis. Please correct me if I misunderstood anything.
- This paper did a poor evaluation of the PDE part. In the PDE surrogate modeling, UNet is not the state-of-the-art model (authors made a wrong and misleading claim about the performance of UNet in PDE modeling) and performs much worse than the Fourier Neural Operator (FNO) [1][2][3]. So I am not convinced that it is useful to only demonstrate the effectiveness of Multi-ResNets compared to UNet in PDE modeling. In addition, no other representative baselines are compared within the NS and shallow water experiments, such as FNO[1], MPPDE[4], FFNO[5], etc.

[1] Li, Zongyi, et al. "Fourier neural operator for parametric partial differential equations." arXiv preprint arXiv:2010.08895 (2020).\
[2] Takamoto, Makoto, et al. "PDEBench: An extensive benchmark for scientific machine learning." Advances in Neural Information Processing Systems 35 (2022): 1596-1611.\
[3] Helwig, Jacob, et al. "Group Equivariant Fourier Neural Operators for Partial Differential Equations." arXiv preprint arXiv:2306.05697 (2023).\
[4] Brandstetter, Johannes, Daniel Worrall, and Max Welling. "Message passing neural PDE solvers." arXiv preprint arXiv:2202.03376 (2022).\
[5] Tran, Alasdair, et al. "Factorized fourier neural operators." arXiv preprint arXiv:2111.13802 (2021).

**Questions:**

Please refer to the weakness part.

**Limitations:**

The limitations of this paper are well discussed.

---

> ### Author Rebuttal · Authors · 2023-08-09
>
> We thank the reviewer for their review, and for highlighting various components and insights of our unified framework as strengths of our work. Below, we would like to focus on the two weaknesses the reviewer mentions.
>
> > “The presentation of the paper is not smooth and clear, which needs significant improvement.”
>
> Your feedback concerning the paper's clarity is invaluable, and we are committed to making necessary revisions. If you have further specific points or sections that you found unclear, we would greatly appreciate further guidance to target those areas effectively.
>
> > “[...] I don’t see how this model can help researchers design specific UNet for different tasks. It’s more like the authors improve UNet on three chosen tasks [...]. Please correct me if I misunderstood anything.”
>
> We value your recognition of our efforts in enhancing U-Nets for several tasks. Regarding your concern, our paper aims to unify the U-Net definition and component roles based on literature. The inclusion of Multi-ResNets is not for introducing a new U-Net architecture per se, but rather as a tool to analyse the encoder's role (Sec 3.1) and validate it empirically (Sec 5.1). This approach empowers researchers to make informed U-Net component choices, in contrast to empirical iteration practised.
>
> In Sec 3, we offer three examples of novel design choices within our framework that incorporate task-specific requirements and constraints into U-Nets. Among them are:
>
> ***U-Nets for complicated geometries:***
> Our framework enables the design of U-Nets over complicated geometries beyond the square, for instance CW-complexes or manifolds. See “B. Diffusions over complicated geometries” in our response to reviewer 7MgH for a complete explanation.
>
> ***U-Nets with constraints:***
> Suppose that we wish to build a U-Net which is guaranteed to output functions of a certain smoothness class and boundary condition. This could be for making a U-Net for PDE surrogate modelling, where we know a priori that the PDE has a weakly differentiable solution with nullified boundary conditions. Then for $W_i$ in our U-Net definition, we can choose a basis of triangular basis functions (Sec 3.2). We encode this within our neural network by selecting a layer of the $D_i$ to model the coefficients of this basis to a given dimension through what our framework and definition propose for selecting the approximation space which is desirable to the problem at hand.
>
> We would appreciate it if you could consider these two examples (alongside the Multi-ResNet)  in response to your comment on designing task-specific U-Nets towards your final evaluation of our work.
>
> > “This paper did a poor evaluation of the PDE part. In the PDE surrogate modeling, UNet is not the state-of-the-art model (authors made a wrong and misleading claim about the performance of UNet in PDE modeling) and performs much worse than the Fourier Neural Operator (FNO) [1][2][3]. [...] In addition, no other representative baselines are compared within the NS and shallow water experiments, such as FNO[1], MPPDE[4], FFNO[5], etc.”
>
> It is important for us to stress that this paper is not about PDE surrogate modelling. This paper is about U-Nets, their design and analysis. In our experiments, we chose PDE surrogate modelling as one of three tasks, because U-Nets are a competitive model candidate for each of them, and often the go-to choice for practitioners. The goal of Sec 5.1 was to verify our theory on the role of the encoder, and indeed provide empirical evidence, including on PDE modelling, that parameters in the encoder may sometimes not be as useful as one might think, if the encoder spaces were chosen suitably—a non-trivial insight enabled by our theory. We believe it would be interesting to benchmark U-Nets and Multi-ResNets against further models such as the ones the reviewer mentions, possibly even on more datasets, yet believe this is beyond the scope of this paper.
>
> We respectfully disagree with the statement that we had made a “wrong and misleading claim about the performance of [the U-Net] in PDE modelling”. In our submission, we have neither claimed that U-Nets outperform FNO, nor have we claimed that U-Nets are ‘the best’ model for PDE modelling at present. Analysing this was not a goal of this paper. Our intent was to highlight U-Nets as a competitive architecture for the tasks at hand, not necessarily as the unequivocal best choice. We regret any misunderstanding that the term "state-of-the-art" may have caused. While sources like PDEBench [2] also reference U-Nets in this manner (“[...] with results for popular state-of-the-art ML models (FNO, U-Net, PINN) [...]” [2]), we'll revise the term to "competitive" in all occurences to prevent any misinterpretations.
>
> That said, we disagree with the reviewer's assessment of FNO's dominance over U-Nets. We believe references [1-3] do not support their claims of clear FNO superiority for various reasons, and we're happy to discuss these papers at length individually during the author-reviewer discussion period. We would further like to refer to the PDEArena benchmark [7], the perhaps most recent, large-scale PDE benchmark which reports U-Nets as competitive, sometimes clearly superior to FNO. In addition, we have run independent, new experiments beyond our submission which underline this on the scale which we report in our submission: in Table 1 in the author rebuttal PDF, we have conducted a comparison of our reported U-Nets with FNO models of similar size. These results match the results in PDEArena by order of magnitude (see Table 2 in author rebuttal PDF), noting that we use slightly different experimental configs. Both report the result that U-Nets outperform FNO. We hence conclude that describing U-Nets as “competitive” for PDE Modelling is an appropriate claim.

---

> > ### Comment · Reviewer_ndAy · 2023-08-18
> >
> > Most of my concerns have been addressed by the author's response. Regarding the performance comparison of FNO and UNet, PDEArena and PDEBench draw the opposite conclusion, so the current benchmark paper on PDE is not a gold standard yet. So it's better for the authors to run PDE baseline models, such as FNO, by themselves and compare with their modified UNet model (after all one study case is PDE modeling, so it is better to not only have UNet result but also include other baseline models.). I understand that this paper is not solely about PDE modeling, and given the fact that my other concerns are addressed, I will increase my score to 5.

---

> > > ### Author Response · Authors · 2023-08-18
> > > **Response to: Official Comment by Reviewer ndAy**
> > >
> > > We thank the reviewer for their suggestions which have improved the quality of our paper, and for raising their score in light of the discussion. In a potential camera-ready version, we will include the experiment we presented in the author rebuttal PDF, where we conducted an independent FNO baseline comparison to U-Nets. In this experiment, we observed that, for a comparable model size and on two datasets (Navier-Stokes, Shallow water), U-Nets outperform FNO in terms of r-MSE (across multiple random seeds), in some cases by an order of magnitude. We will also include a more thorough literature review on the topic.

---

### Official Review · Reviewer_Q8QJ · 2023-06-30

**Soundness:** 4 excellent
**Presentation:** 3 good
**Contribution:** 3 good
**Rating:** 7
**Confidence:** 2

**Summary:**

The paper presents a unified framework for U-net design, by generalizing the overall structure of U-nets into different components. The framework is then investigated from a theoretical point of view. At its core, the paper presents Multi-Resnets which is a novel class of U-nets, which is then tested rigorously in the experimental section, in relation to generative modelling, PDE modelling and image segmentation.

**Strengths:**

It is always great to see papers that take a step back and tries to unify different work into a framework, to conceptualize what is up and down in a certain field. In that sense this paper success in doing this. In particular, I think sections such as 5.1 are some of the stronger sections of the paper because they conceptualize what the role of the encoder is.

Originality:
The framework presented in the paper seems to be original compared to other work.

Quality:
The paper seems to be of a high quality, with both theoretical results to backup the claims of the paper but also empirical evidence to support. This in especially true, when the appendix of the paper is taken into account as it provide much more information, especially on the experimental section of the paper.

Clarity:
The paper is fairly clear in what it is trying to achieve, however since there are a lot of moving parts to the paper the overall structure sometimes suffer from this and the overall story become a bit "muddy".

Significance:
While the conceptualization of U-nets indeed is refreshing, I doubt this paper will have a huge impact on domains that use U-nets as most established fields seems to build incrementally upon each others instead of redesigning architectures. That said, only time will tell.

**Weaknesses:**

I did not in particular find any weaknesses of the paper.

**Questions:**

Maybe out of scope for the paper, but let me ask anyway:
The original reason for resnet being introduced was to solve the problem of vanishing gradients. Do you see a connection between this problem and your work?



**Limitations:**

The authors provide a lengthy explanation of what the limitations of the framework is in the appendix of the paper, however I think the authors should have definitely included this in the main paper.
To me, while the framework generalizes U-Nets it does not necessarily guarantee that the design process becomes easier, as the authors also touch on L162-166. This indeed does seem to be a limitation of the framework, when the initial goal was to "..designing and analysing general U-Net architectures".

---

> ### Author Rebuttal · Authors · 2023-08-09
>
> We thank the reviewer for their positive feedback on our work. We are delighted that you appreciate the approach of "taking a step back" in our research and our analysis of the role of the encoder in U-Nets (Sec 5.1), as well as the originality and quality of our theory, experiments, and Appendix. Your support is greatly valued.
>
> We would like to respond to your review below. We appreciate the opportunity to address any points you have raised and further engage in constructive discussions about our research.
>
> > “It is always great to see papers that take a step back and tries to unify different work into a framework […] In that sense this paper success in doing this. Significance: [...] I doubt this paper will have a huge impact on domains that use U-nets as most established fields seems to build incrementally upon each others instead of redesigning architectures. That said, only time will tell.”
>
> On significance, we concur that many fields evolve through incremental advancements, but the very act of 'taking a step back' has enabled us to unearth several novel insights. For instance, our exploration into the role of the encoder in U-Nets led to the design of Multi-ResNets. Furthermore, our findings on U-Nets' self-similarity structure, high-resolution scaling behaviour, and the capability to integrate function constraints (like PDE boundary conditions) directly into the U-Net architecture are pivotal. We've also ventured into designing U-Nets for complex geometries, such as manifolds and spheres in particular.
>
> One of our aspirations with this paper is to encourage the exchange of U-Net design choices across different domains. As an example, the recent PDEArena benchmark [7] demonstrated the high applicability of U-Nets popular in computer vision to PDE surrogate modelling. Notably, in [7], U-Nets are competitive with other architectures (such as Fourier Neural Operators (FNO)) in that field. We aim to foster a common language for this go-to, widely-used architecture design and believe that many more advancements can be made through such cross-domain efforts based on a unified framework. Our hope is that this work will serve as a catalyst for the exchange of ideas and foster advancements in U-Net design, ultimately benefiting a wide range of applications and research areas.
>
> > “Maybe out of scope for the paper, but let me ask anyway: The original reason for resnet being introduced was to solve the problem of vanishing gradients. Do you see a connection between this problem and your work?”
>
> We thank the reviewer for this intriguing question. We have not extensively analysed the connection between our work and vanishing gradients, which have indeed been addressed in practice through the use of residual and skip connections. However, we firmly believe that there are fascinating properties on the vanishing gradient problem to be explored now that a U-Net is defined (see Def 1). Specifically, we are interested in defining a ‘stable U-Net’ (akin to similar work on stable ResNets [8]) by understanding the minimal requirements on the choice of $E_i$ and $D_i$ so that the model is guaranteed to have gradients bounded away from zero and infinity, independent of width and depth of the network. We acknowledge the importance of this area of research and look forward to further investigations into the stability properties of U-Nets.
>
> On another note, our work was indeed greatly motivated by the seminal ResNet paper [6] through the idea of *preconditioning*, which is the core design principle of ResNets and U-Nets alike (also visualised in Figure 2 of our submission). This ultimately led us to the insight of a conjugacy between ResNets and U-Nets (see Prop 1).
>
> > “The authors provide a lengthy explanation of what the limitations of the framework is in the appendix of the paper, however I think the authors should have definitely included this in the main paper.”
>
> We agree with the reviewer, the limitations discussed in Appendix A are important and useful to be shown in the main text. Following the reviewer’s suggestion, we will include this discussion in Sec 6 (Conclusion) of the main paper as part of the additional page in a potential camera-ready version of this manuscript.
>
> > “[...] To me, while the framework generalizes U-Nets it does not necessarily guarantee that the design process becomes easier [...]. This indeed does seem to be a limitation of the framework [...]”
>
> Our framework offers a decisive advantage in the design process of U-Nets, allowing successful design choices from one area to be effectively applied in another. One illustrative example is the construction of generative models over triangulated spaces, which has significant implications for modelling functions over complicated geometries.
>
> Specifically, when faced with the task of designing a U-Net for a diffusion model with a favourable inductive bias on a triangulated complicated geometry, we have demonstrated in Theorem 2 that utilising a Haar wavelet decomposition of the geometry naturally induces a desired inductive bias for generative modelling. By opting for a Haar decomposition of the triangulated domain, the selection of spaces $V_i$ and $W_i$ is immediately defined, with Theorem 2 further specifying $P_i$. In this sense, our framework makes designing U-Nets ‘*easier*’ by providing a theoretical framework in which design choices for each of the components of a U-Net can be made based on theoretical insight and by incorporating problem constraints, rather than based on an empirically-driven ‘trial-and-error process’. Notably, our framework also extends the applicability of Theorem 2 to a broader range of geometries, enhancing the generality of its theoretical contributions.
>
> In light of the reviewer's input, we recognise the need to make these connections more apparent in a potential camera-ready manuscript. We welcome any further suggestions on how to improve the communication of these ideas in our manuscript.

---

> > ### Comment · Reviewer_Q8QJ · 2023-08-15
> > **Thanks for your answers**
> >
> > I thank the authors for their lengthy answer, to not only my own review but also my fellow authors. Especially, thanks for being very honest in your answers being aware of your papers weaknesses and still defending the strengths of your paper. The mentioned improvements to the potential camera-ready manuscript sounds like great improvements to the paper.
> > I have no further questions and comments for the paper. I will keep my score at 7 as it is already in higher end based on the other reviews.

---

> > > ### Author Response · Authors · 2023-08-16
> > > **Response to: "Thanks for your answers"**
> > >
> > > We thank the reviewer for their useful questions and remarks which have helped to improve the quality of our submission towards a potential camera-ready version. We are grateful for the efforts in carefully reviewing our work. Lastly, we thank the reviewer for maintaining the high score and viewing to accept our article.

---

### Official Review · Reviewer_7MgH · 2023-07-02

**Soundness:** 4 excellent
**Presentation:** 4 excellent
**Contribution:** 3 good
**Rating:** 6
**Confidence:** 4

**Summary:**

The work proposes a unified mathematical framework to analyze and design U-Net. Authors highlight the importance of preconditioning and provides several highlights to design unet. Authors propose a new parameter-free encoder based on wavelet space.

**Strengths:**

1. Sec 2 presents a unified mathematical framework for U-Unets.
2. The discussion of precondition parts and how to design various precondition for different tasks is insightful, such as sec 3.2, 3.3.
3. Authors also conduct various experiments to support the their claims and highlight the importance of precondition and network design. Some experiments are quite interesting to me, such as Sec 5.1 and 5.3.

**Weaknesses:**

1. Though it shows improvement in other tasks, there are minimal improvements in diffusion model tasks. Authors should report performance of Multi-ResNet for diffusion models and compare it with existing U-Nets used widely in diffusion models.
2. I would like authors to comments on how to design Unet for diffusion model so that it can recover high frequency details in high resolution image generation task. The average pooling will filter high frequency noise, but also filter the high frequency data information.

**Questions:**

See weakness

**Limitations:**

See weakness

---

> ### Author Rebuttal · Authors · 2023-08-08
>
> We are glad to hear the reviewer appreciated our submission.
>
> > “Though it shows improvement in other tasks, there are minimal improvements in diffusion model tasks. Authors should report performance of Multi-ResNet for diffusion models and compare it with existing U-Nets [...]”
>
> It is important for us to stress that generative modelling with diffusion models is only one of the three tasks (besides PDE surrogate modelling and image segmentation) which we tackle to verify hypotheses and ideas derived from our unified U-Net framework. Diffusion models are hence not the core focus of this work, it is rather one (currently very popular) area of research where the U-Net architecture produces strong results.
>
> Our key findings are irrespective of the choice of neural network one uses to parameterise the NNs $\mathcal{E}$ and $\mathcal{D}$ (with the caveat that Multi-ResNets require a ResNet decoder). Comparing different U-Net architecture choices for diffusion models is hence a very interesting direction which should definitely be pursued, particularly in light of this paper, and we thank the reviewer for this idea. We yet believe it is not essential to support the contributions of this work.
>
> Besides, we would like to briefly recap 3 key improvements our unified framework provides *specifically for diffusion models*:
>
> A. Theoretical understanding:  While not an improvement for diffusion models in their design or performance, we believe Sec 4 is a crucial improvement and contribution for understanding the understudied success of U-Nets in diffusion models, especially on images. To the best of our knowledge, Theorem 2 for the first time proves why a U-Net with average pooling as their projection operators $\mathcal{P}$, and regardless of the choice of operators $\mathcal{E}$ and $\mathcal{D}$ is a natural inductive bias for data under a forward noising process of a diffusion model. Specifically, Theorem 2 shows that – possibly in contrast to the intuition prior to this submission – that not all frequencies of an image are affected equally by the forward noising process.
>
> B. Diffusions over complicated geometries: Our framework enables the design of U-Nets over complicated geometries beyond the square, for instance CW-complexes or manifolds. This can for instance be used to design neural network architecture for diffusion models on the manifolds merely by encodes the topology of the data into U-Net itself. As a proof-of-concept with U-Nets over a Haar wavelet basis and on a triangular domain (see Sec 3.3 and Sec 5.3), and discuss how this can be extended via triangulations to design a U-net for a sphere (see Appendix A.2) and other triangulated spaces.
>
> This may be particularly successful as at present, such work often leverages simple, fully-connected networks as their neural architecture. For instance in [4] (also mentioned by reviewer go4z), an award-winning diffusion models paper from last year’s NeurIPS, which extends diffusion models to Riemannian manifolds, the authors exclusively use fully-connected neural networks in their experiments (see appendix O in [4], “Architecture”). Our work is hence complementary to such work, in the sense that it enables a large, previously unrecognised potential in the neural architecture.
>
> C. Multi-resolution training and sampling: As we experimentally demonstrate in Sec 5.2, the identification of the self-similarity structure of a U-Net in Sec 2 enables us to naturally train and sample from diffusion models on multiple resolutions via Algorithm 1 (see Appendix B.2), by training $\mathcal{U}$ on resolution $i-1$, then training $U_i$ while preconditioning on $U_{i-1}$. Two advantages arise: First, if data is available on a higher resolution $i$, we can reuse the U-Net on a lower resolution $U_{i-1}$ in a principled way, i.e. by preserving the self-similarity structure of the neural network. Second, we can checkpoint sampling models when prototyping, look at and possibly measure the quality of lower-resolution samples (see Fig. 6) first, which may be indicative for the quality of the higher resolution.
>
> > “I would like authors to comments on how to design Unet for diffusion model so that it can recover high frequency details [...].”
>
> We find the reviewer's question intriguing. While we are uncertain we believe that the reviewer's inquiry pertains to super-resolution applications. Super-resolution techniques hinge on identifying long-range dependencies within image structures to predict higher resolutions. For the purposes of our response, we will assume that our image super-resolution algorithm is trained using high-resolution training data.
>
> First, we require the U-Net model to accommodate images of varying resolutions. The essential design requirement is the U-Net's ability to accept images of different resolutions. The original U-Net lacks this capability. However, our Multi-ResNet inherently addresses this issue, enabling the U-Net to handle input images with varying resolutions seamlessly. This is because the encoder has no learnable parameters, and in essence, makes various projections onto varying resolution spaces of the original image.
>
> Suppose now that we are given data on resolution $V_{i+1}$ (a higher resolution than $V_i$) and train a diffusion model with a Multi-ResNet on this data. Then we are given data of resolution $V_i$ and asked to produce a super resolution version of this image. What is important here is, as the reviewer says, that in this setup, any image of arbitrary resolution can be effortlessly embedded into one of the resolution input spaces, denoted as $V_i$, through interpolation. This elegantly extends the U-Net model to support resolutions of any scale, making it a versatile and straightforward approach applicable even to infinite resolution scenarios. This model may not perform well, but being the simplest U-Net model that is adaptive to this context, it forms a good baseline method to ablate from.

---

### Author Rebuttal · Authors · 2023-08-09

We thank the reviewers for their valuable and insightful comments and questions. Motivated by the reviews, we make several changes towards a potential camera-ready version of the submission, using the one additional page, which we list below.

### Summary of key changes towards a camera-ready version

* A note on how to extend our framework to arbitrary channel dimensions using the Kronecker product in Sec 2.1.
* A paragraph on the limitations of our work in Sec 6 (Conclusion), based on the respective section in Appendix A.
* Algorithm 1 and Appendix A.3 as a condensed version, now in Sections 5.2 and 3.2.

### Response on other points from reviewer go4z:

> “Algorithm 1 for staged training should be in the core paper [...] a U-Net that guarantees boundary conditions should be in the core paper.”

We will address this clarity issue by incorporating practical approaches for guaranteed boundary conditions with a U-Net, adding Algorithm 1 to Sec 5.2 and a condensed version of Appendix A.3 in Sec 3.2 as per the reviewer's insightful feedback.

> The formal definition of the U-Net is hard to grasp at first read. There should be a correspondence between the elements introduced and the typical 2D image segmentation U-Net architecture.

We agree and will provide a figure further illustrating Definition 1 in a potential camera-ready version.

> To me preconditioning means finding a matrix close to the Hessian inverse in an optimisation procedure [...] Here it seems that preconditioning is more used to describe a good initialisation [...]

We acknowledge the reviewer's observation that “preconditioning” can refer to finding a matrix close to the Hessian inverse for an optimisation procedure. However, we would like to emphasise that “preconditioning” is a term which implies different meanings across various contexts in mathematics. The seminal ResNet paper uses the word ‘precondition’ to indicate ‘initialisation to the identity: “In real cases, it is unlikely that identity mappings are optimal, but our reformulation may help to precondition the problem [6].” As this U-Net work has strong ties with ResNets (e.g. their conjugacy in Prop 1), we considered using it in the same way seems appropriate. We appreciate the reviewer's feedback, which allows us to enhance the clarity and quality of our work.

> “[...] how the decoder should be designed in order to achieve these subspaces.”

We value the reviewer's input on improving our manuscript quality. The decoder's design corresponds to predicting linear combinations of subspace $W_i$ basis vectors. Haar wavelets use a pixel grid as a valid basis, Fourier layers for instance transform into linear combinations of Fourier modes of certain frequencies. Generally, $w_i \in W_i$ is expressed as $w_i = \sum c_i e_i$ with basis vectors $e_i$. The neural network predicts coefficients $c_i$ for a fixed data basis. We'll explicitly address this in the paper update and appreciate the reviewer's observation which helps clarify this point.

> “It also explains why in practice, encoders are often chosen significantly smaller than the decoder [26]’. I am not sure about this fact [...]”

[26] indeed focuses on hierarchical VAEs, which however employ U-Net-style architectures, extensively discussed in [5] (Appendix B.2). Notably, [26] presents an interesting scenario with a smaller encoder than the decoder, possibly explained by our framework. We'll revise 'often' to 'in some cases' to better reflect varying encoder-decoder choices in diffusion models. Thank you for this remark.

> “What is meant by self-similarity?”

We appreciate the reviewer's input on terminology clarity. "Self-similarity" in our context refers to refining a square image into smaller squares to establish nested approximation spaces ${W_i}$ for our U-Net. This enables the U-Nets' adaptability to recursively refined geometries. While acknowledging the non-standard nature of this term, we'll enhance precision in Sec 2.2 to address this feedback. Thank you for helping refine this aspect of our work.

> “What is it [GNet] and does it fit with the experiments reported in the paper?”

We appreciate the reviewer's notice of a code typo. "GNet" was our code name for the ‘general’ U-Net class. This term was specific to initial diffusion experiments, and we will correct it in the potential camera-ready submission.

> “U-Net acts on feature spaces, i.e. spaces with much more feature channels than the image itself, how would you take that into account?”

Our framework easily extends to multiple channels by utilising the kronecker product to expand W_i to the desired channel count in a U-Net. In a potential camera-ready version, we'll add a note in Sec 2.1 and a dedicated appendix section for the construction details. Thank you for this clarifying comment.

### References:

[1-3]: see Reviewer ndAy

[4] De Bortoli, V., et al. 2022. Riemannian score-based generative modelling. Advances in Neural Information Processing Systems, 35, pp.2406-2422.

[5] Falck, F. et al., 2022. A Multi-Resolution Framework for U-Nets with Applications to Hierarchical VAEs. Advances in Neural Information Processing Systems, 35, pp.15529-15544.

[6] He, K., et al., 2016. Deep residual learning for image recognition. In Proceedings of the IEEE conference on computer vision and pattern recognition (pp. 770-778).

[7] Gupta, J.K. and Brandstetter, J., 2022. Towards multi-spatiotemporal-scale generalized pde modeling. arXiv preprint arXiv:2209.15616.

[8] Hayou, S., et al., 2021, March. Stable resnet. In International Conference on Artificial Intelligence and Statistics (pp. 1324-1332). PMLR.

[9 Hoogeboom, E.,  et al., 2023. simple diffusion: End-to-end diffusion for high resolution images. arXiv preprint arXiv:2301.11093.

---

### Decision · Program_Chairs · 2023-09-21

**Decision:**

Accept (poster)

**Comment:**

All reviewers agreed that given the success of U-Nets in applications like diffusion modeling and PDEs, the paper does a compelling analysis of different choices in its design space and how it's domain of usefulness can be expanded to more complex geometries. While one reviewer disputed whether U-Nets are superior to FNOs for PDE modeling, this is tangential to the paper's primary focus. Nonetheless, U-Nets have certainly proven to be an important baseline in PDE modeling contexts too and given interest in efficient architectures from the community, would help others in the exploration of this design space.